# Occupation-Induced Fatigue and Impacts on Emergency First Responders: A Systematic Review

**DOI:** 10.3390/ijerph20227055

**Published:** 2023-11-12

**Authors:** Graham Marvin, Ben Schram, Robin Orr, Elisa F. D. Canetti

**Affiliations:** Tactical Research Unit, Faculty of Health Sciences & Medicine, Bond University, Robina, QLD 4226, Australia

**Keywords:** firefighters, paramedics, fatigue, safety, injury, function

## Abstract

Fatigue in emergency first responders (EFRs) is known to affect performance abilities and safety outcomes for both patients and EFRs. The primary aim of this review was to determine the main contributors to occupation-induced fatigue in EFRs and its subsequent impacts. Following the PRIMSA checklist, academic databases (Medline, Embase, CINAHL, and SPORTDiscus) were searched using key terms with results subjected to inclusion and exclusion criteria. Populations of interest were firefighters, paramedics, or emergency call centre personnel. Of the 5633 records identified, 43 studies, which reported on 186 unique measures from a total population of 6373 participants, informed the review. Synthesis revealed fatigue was caused by lack of sleep during the shift and consistent poor sleep quality which negatively impacted cognitive function, alertness, and physical and mental health while increasing safety-compromising behaviours and injuries. Both subjective and objective assessments of fatigue are necessary for effective risk management in EFRs. EFRs that are consistently fatigued are at a greater risk of poor physical and mental health, reduced cognitive function, and increased injuries. No studies reported on fatigue in emergency call centre personnel, highlighting a literature gap. Funding was provided by the Australian Capital Territory Emergency Services Agency. Preregistration was filed in OSF: osf.io/26f3s.

## 1. Introduction

Emergency first responders (EFRs), such as firefighters and paramedics, respond to various emergencies and critical events to protect the safety, property, and lives of citizens [1,2]. The occupational requirements often include working long hours, sleepless nights, and high-stake job demands [3,4,5]. In the long term, these demands can cause persistent elevated levels of occupational fatigue [6], which is known to be a pervasive and complex problem impacting occupational performance [7,8].

When EFRs combine occupational fatigue with the known extensive physical and cognitive requirements of the job [9], the downstream impacts are, unsurprisingly, cognitive [10,11,12,13] and physical performance deficits [8,12,14]. In EFRs, cognitive deficits may be a result of reduced cerebral blood flow after sleep deprivation [15]. For example, after one night of being deprived of sleep, EFRs displayed increased reaction times and decreased information processing speeds and visual–motor coordination [16]. A decrease in cognitive function can be detrimental to EFRs due to delayed reaction times to unexpected events [17]. Meanwhile, physical deficits can result from reductions in neuromuscular transmission leading to an inability to maintain physical performance and thus increased injury risk [7,18]. Injuries can present as slips, trips, and falls (the most common injury mechanism in firefighters [19]) and lifting related overexertion injuries (common in paramedics [20]) serve as examples. Generally, occupational fatigue increases the risk of accidents while compromising personal safety and patient safety (e.g., from procedural/medication errors [5]) and can increase injury risk by 1.6-fold [21]. Unfortunately, EFR occupational injuries are problematic worldwide.

In South Korea, firefighters sustained 116 injury cases per 1000 firefighters during 2015 [22]. The injury rates caused 1044 days of absence per 1000 firefighters, which equated to an average of 2.8 days off per injury [22]. In Australia, firefighters sustained 100 injuries per 1000 workers, which equated to losing 721 weeks from work and was three times higher than all other occupations [23]. Similarly, in United States (US) firefighters, the non-fatal injury average from 2016–2020 totalled 23,610 [24], while the total annual cost of injuries ranged from USD 1.6–5.9 billion [25]. Moreover, in US paramedics, the injury rate is 54 per 1000 workers and is four times higher than national average for all occupations [26]. Additionally, in Australia, paramedics have the highest rate of injury claims of any occupation at 150 per 1000 workers [23]. Furthermore, paramedics report time lost due to injuries which is seven times higher than all other occupations in Australia [27]. Although no monetary values have been established for the EFR population, Hillman et al. [28] reported that Australia’s total workplace injuries resulting from sleep deprivation, which runs rampant across EFRs [29,30], are estimated to cost roughly AUD 300 million annually. Considering this, sleep deprivation is just one contributor to occupational fatigue [31] and other contributors may exist that play a role in cognitive and physical fatigue within EFRs. 

Determining what contributes to occupational fatigue can be difficult due to the fact that fatigue-related impairments have multiple causes and affects different skills [32]. Given the high individual and organisational cost of fatigue, the primary aim of this review was to determine the main contributors to occupation-induced fatigue in EFRs and their subsequent impacts. The secondary aim was to elucidate and discuss relevant fatigue-mitigation strategies which could be implemented in EFR organisations for a safer workplace.

## 2. Methods

The Preferred Reporting Items for Systematic Reviews and Meta-Analyses (PRISMA) checklist guided this review [33]. The protocol was registered in the Open Science Framework and was completed prior to the search being conducted (osf.io/26f3s).

### 2.1. Eligibility Criteria

Reports that were peer-reviewed, original research, and translatable to English were considered for inclusion in this study. The population of interest was firefighters, paramedics, and emergency call centre personnel of any sex. The outcomes of interest were contributors to fatigue, cognitive or physical, that fell within the definition by the International Civil Aviation Organisation (ICAO): “a physiological state of reduced mental or physical performance capability resulting from sleep loss, extended wakefulness, circadian phase, and/or workload (mental and/or physical activity) that can impair a person’s alertness and ability to perform safety-related operational duties” [32,34]. Included studies were restricted to the last ten years to keep the most up-to-date information and to ensure that the tasks performed by EFRs were relevant in the current environment. Reports were excluded if the population contained data pooled with other occupations, including nurses, physicians, and police, for which the population data of interest could not be extracted. 

### 2.2. Information Sources

Potentially relevant literature was identified using a 3-step approach. First, a preliminary search was conducted in two electronic databases pertinent to the topic: Medline and Embase. Using three relevant studies (PMID: 22023164, 26840323, 33000071) identified in the preliminary search, SR-Accelerator [35] was used to optimise search terms to ensure the most concise and effective search strategy. A second comprehensive search strategy was then developed by including all identified keywords and MeSH terms by the author (GM) with the aid of a university librarian. Four databases (Medline, Embase, CINAHL, and SPORTDiscus) were searched using the final search strategy, and the records identified were downloaded and finalised on 29 November 2022. The search results were imported into EndNote (X9 Clarivate Analytics), where duplicates were removed. Using a backward snowballing approach, the reference lists of all included studies were scanned to identify potential studies to be included. Any studies identified through this process were subject to the same inclusion and exclusion criteria.

### 2.3. Search Strategy and Selection

Search terms used were based on their availability within their respective databases and were adjusted to ensure that concepts of fatigue and emergency personnel were captured. The search terms used in each database are detailed in Table 1. Where available, filters were applied within each database limiting the search to the last ten years as per the eligibility criteria. After removing duplicates, the principal investigator (GM) independently screened the articles based on the titles and abstracts. No automation tools were used. Two other authors (BS and RO) assisted with any discrepancies. The full texts of the remaining reports were retrieved and screened for eligibility. Reports that did not meet the eligibility criteria were removed. Per PRISMA checklist guidelines [33], a flow diagram outlining the report selection process was created.

### 2.4. Outcome Measures and Synthesis Methods

Outcome measures of interest were those that assessed fatigue, whether it was cognitive or physical, with both subjective and objective measures retrieved. Data pertaining to the author, year of publication, demographic details of the population, and study type were also extracted and tabulated. Results extracted from studies included the total number of participants, measures of fatigue, length of study, findings, and key summary points. The summation of the studies was then graded based on the National Health and Medical Research Council (NHMRC). 

Any item that fell under the umbrella of the ICAO definition of fatigue was categorised as a direct contributor to fatigue. Additional items that were within the same study but fell outside the ICAO definition of fatigue were classified as an indirect contributor to fatigue. Direct contributors to fatigue were categorised based on the definition with eight overarching themes: sleep practices, fatigue questionnaires, physiological measurements, cognitive reaction tests, self-rated measures, physical activity, workload and safety, and balance and strength. The themes created in the indirect contributor category were mental health and recovery practices aimed at reducing fatigue. 

### 2.5. Risk of Bias Assessment

The Joanna Briggs Institute (JBI) was used to determine the risk of bias and internal validity of the studies included [36]. The risk of bias is a preferred method over methodological quality to grade reviews according to the PRMISA [37]. The JBI assessment is a standard tool that is widely accepted in systematic reviews [38] and commonly used for cohort, case-control, cross-sectional, and case-report studies [39]; the study types included in this review. To minimise bias, studies were assessed for risk of bias by two authors (GM and BS) independently. Interrater agreement was assessed using a Cohen’s Kappa coefficient (k) calculation. The JBI for cross-sectional studies contained 11 questions, for qualitative studies 10 questions, and for cohort and case-series studies, eight questions, respectively. The result of each study on the JBI that was answered ‘yes’ was indicated with a ‘+’, any ‘no’ was given ‘-’, and any question that was not applicable was given a ‘N/A’. Scores were given in percentages based on the total numbers of each item included, divided by the total and rounded to the nearest whole number. Scores were assigned as fair agreement (k = 0.21–0.40), moderate (k = 0.41–0.60), substantial (k = 0.61–0.80), or almost perfect (k = 0.81–0.99) [40]. Disagreements were discussed with a third author (RO) until a consensus was reached. The risk of bias of each included study is presented in tables, grouped by study design, and presented in the results section as recommended by the PRISMA guidelines [37]. 

### 2.6. Effect Measures

The included studies used various measures to assess fatigue and could not be distilled into one type of outcome. Each individual study was tabulated with their outcome and effect measure reported. These effect measures range from *p*-values, risk ratios (RR), confidence intervals (CI), standardised mean differences (SMD), correlation coefficients (r and r^2^), beta values (β), omega squared (ω^2^), F-values (F), eta squared values (η^2^), log-transformed root mean square of successive R-R intervals (lnRMSSD), chi-squared distributions (X^2^(2)), and odds ratios (OR). Alpha levels of significance (*p*-values) ranged from <0.05 to <0.001 and correlation coefficients (+/−0.30) were based on the study authors’ chosen values of significance.

## 3. Results

Of the initial 5633 records identified, 95 eligible studies were screened by full text. Forty-two studies met the eligibility criteria [41,42,43,44,45,46,47,48,49,50,51,52,53,54,55,56,57,58,59,60,61,62,63,64,65,66,67,68,69,70,71,72,73,74,75,76,77,78,79,80,81,82] with the addition of one study through the identification of backward snowballing [83] (Figure 1). Fifty-three text reports were excluded, with reasons for exclusion documented in Figure 1. The most common reasons for being excluded were a failure to meet the definition of fatigue (n = 24) and a lack of full text availability (i.e., only published abstracts; n = 19). Another 10 reports appeared to meet the inclusion criteria but were ultimately excluded due to the population of interest being mixed with other populations such as physicians, police, or pilots for which the individual data could not be extracted [84,85,86,87,88,89,90,91].

Of the 43 included studies, 25 were cohort [42,43,46,49,51,52,54,56,57,59,61,62,63,65,66,67,72,74,75,78,79,80,81,82,83], 15 were cross-sectional [41,45,47,48,50,53,58,60,64,69,71,73,76,77,92], two were qualitative [44,68], and one was a case report [70]. 

Seventeen of the included studies were published in Australia [41,43,44,49,50,60,61,68,69,73,75,76,78,79,80,81,82], seven from the United States [42,46,51,52,54,70,71], six from Canada [47,48,56,57,66,83], two from Poland [62,67], Taiwan [64,93], and South Korea [58,63], and one each from Brazil [45], Finland [59], France [65], Iran [53], Italy [72], Japan [77], and Spain [74].

These studies yielded 6373 participants, comprising 5159 males, 997 females, and 217 participants with no sex stated. Nine studies only included a male population [46,51,62,65,66,68,70,72,77,92]. Studies that included males and females found no differences in fatigue and performance between the two sexes [41,47,50,60,64,76]. 

Job experience ranged from 0 to 48 years and was not reported in 18 studies [41,43,45,47,49,52,59,60,61,62,63,66,70,73,74,75,77]. In total, the participants’ ages ranged from 19 to 68 years. Populations were reported based on how each author reported job titles, with several studies containing multiple populations. The populations consisted of salaried firefighters [42,46,50,51,52,54,58,59,62,63,65,67,68,72,74,81,82,92], on-call firefighters [43,44,49,68,81,82], wildland firefighters [56,57,66,78,79,80,83], military firefighters [45], rural firefighters [75], petrochemical firefighters [53], emergency medical technician/basic paramedics [41,47,48,60,61,67,69,70,71,73,76,77], and emergency medical technician paramedic intermediates/advanced care paramedics [64]. Populations were generally evenly split between firefighters (n = 3271) and paramedics/EMTs (n = 3102). Although emergency call centre personnel were within the search strategy, no studies reported on this population. Individual demographics of the included studies are detailed in Table 2.

### 3.1. Risk of Bias

Overall, the included studies had a low risk of bias with an average score of 80% and a moderate level of agreement (k = 0.69) [40] between the two reviewers (GM and BS). After discussion, a second analysis revealed a 100% agreement between reviewers. The majority of the disagreements were with Question 5 in the cohort studies and Question 6 in the cross-sectional studies pertaining to confounding variables. After conferring, the reviewers (GM and BS) agreed that if there was at least one mention of a confounding variable the question was marked ‘yes’. This still led to 80% of the cohort and 53% of the cross-sectional studies making no mention of confounding variables. Similarly, pertaining to Question 6, 25% of the cohort studies made no mention whether participants were free of fatigue at the start of the study. Table 3 represents a visual risk of bias assessment scoring from the JBI organised by study type.

### 3.2. Outcome Measures of Fatigue

A total of 186 outcome measures were mentioned in the included studies, and outlined in Table 4. Of these, only 154 (83%) were further analysed by the included studies. The difference between the two numbers is based on how studies presented their data, if there were any findings, or whether data applied to the primary aim. The most common outcome measure used was the Pittsburgh Sleep Quality Index (PSQI), which was reported in nine studies [41,45,60,61,67,70,71,76,92]. Study design, methodology, and assessment of fatigue was very heterogeneous among the included studies, precluding a meta-analysis which allowed the various areas and data to be addressed individually [93].

### 3.3. Direct Contributors

For ease of reporting, the 136 outcome measures that were used to assess the direct contributors to fatigue were collapsed into eight categories of key general measures. These being, sleep practices, fatigue questionnaires and interviews, physiological measurements, cognitive reaction tests, self-rated measures, physical activity, workload and safety, and balance and strength. Each of these are discussed below.

#### 3.3.1. Sleep Practices

Sleep practices were the most common measurement with 43 various outcomes assessed based on 24 studies [41,43,45,52,55,56,57,58,60,61,63,64,65,66,67,70,71,76,78,79,80,81,82,83]. A variety of subjective and objective measures were used to evaluate sleep. Subjective measures included the Pittsburgh Sleep Quality Index Sleep (PSQI) [41,45,60,61,67,70,71,76,92], the Epworth Sleepiness Scale (ESS) [60,64,67,70,76], sleep diaries [56,57,78,79,83], the Insomnia Severity Index [60,63], the Berlin Questionnaire for sleep apnoea [60,76], the Karolinska Sleepiness Scale [61,67], sleep location [78,79], the Ullanlinna Narcolepsy Scale [60], and the Spiegel Sleep Quality Perception Questionnaire [65]. Wearable devices were used to collect objective measures, such as total sleep time, sleep efficiency, sleep latency, and wake after sleep onset [52,56,57,58,61,65,66,78,79,83] while four studies utilised polysomnography [43,80,81,82].

The total amount of sleep was consistently of poor quality [41,45,58,60,61,65,67,70,71,76,92] and, at the same time, regularly being below the recommended guideline amount of 7–9 h a night [56,57,61,66,67,79,82,83]. Poor quality sleep resulted in regular daytime sleepiness [60,64,70,76], as well as excessive sleepiness levels [64,70,76], with reports of falling asleep at the wheel [73,76,79]. These findings were exacerbated for those suffering from insomnia [60,63], narcolepsy [60], and sleep apnoea [60,76]. Unfortunately, sleepiness levels did not improve with two days of rest [56], were worse during the day following a night shift [61], when compared to controls [58], and continued in line with deployment length [57]. Conversely, two authors found firefighters’ sleep was unaffected on workdays compared to non-work days [52], and when planned burn operation shifts lasted less than 12 h [78]. Interestingly, sleep efficiency was increased for those following a fast rotating 6-day shift compared to those in a 21-day shift cycle [58].

#### 3.3.2. Fatigue Questionnaires and Interviews

Twenty-two studies assessed fatigue based on 22 different questionnaires [41,42,45,47,48,49,53,54,60,61,66,70,71,73,76,77,78,79,83], interview-based studies [44,68], and open answer response survey based on fatigue [69]. The most common subjective fatigue measures were the Chalder Fatigue Scale (CFS) [47,48,70,71,73], Samn–Perelli Fatigue Scale (SPFS) [49,61,78,79], and the Occupational Fatigue Exhaustion Recovery Scale (OFERS) [54,70]. Other studies included the Standard Shiftwork Index-Chronic Fatigue Scale (SSI-CFS) [41], Bipolar Fatigue Evaluation Questionnaire [45], the Ecological Momentary Assessment [42], Fatigue Severity Scale [60], Circadian Alertness Simulator [83], the Sleep, Fatigue, and Alertness Behaviour [70], the Multidimensional Fatigue Inventory [53], an interview format fatigue assessment [44,68], one closed ended question about fatigue [77], and an open answer format based on beliefs of fatigue [69].

Both short-term fatigue [42,45,49,60,61,77,78,79,83] and sustained fatigue [47,48,70,71,73] were identified in the majority of participants. However, only one study reported low levels of fatigue in its petrochemical firefighters [53]. Interestingly, eight studies found that not only did fatigue levels increase while on shift [42,44,45,68,69,73,78,79], but stayed elevated the following day after the shift [61,68].

In an open-response survey, significant contributors to fatigue were reported as working time (night shift), sleep (insufficient sleep), workload, health and well-being, work–life balance, and environment [69]. In the qualitative study, interviewers asked EFRs how they believe fatigue physically manifests and responses were lapses in eye–hand coordination, cognitive effects including communication problems and difficulty with decision making, loss of situational awareness (e.g., such as tunnel vision or disengaging), and poor memory recall (e.g., forgetting training procedures) [44]. In one study [54], short-term and long-term occupational fatigue was found to increase with age, which was a predictor of poorer inter-shift recovery. In contrast, the SSI-CFS found no association with age and fatigue but was strongly associated with depression, anxiety, and stress [41]. Fatigue levels during a shift were reduced when shift patterns changed from a 24-h to an 8-h shift [70].

#### 3.3.3. Physiological Measurements

Twenty-one unique physiological measurements were conducted based on eight studies [46,57,59,61,74,80,81,82]. Measures included heart rate [46,59,65,74,80,81,82], core body temperature [74,80,82], heart rate variability (HRV) [57,59,65], galvanic skin response (changes in sweat gland activity) [61], maximum oxygen uptake (VO_2_Max) [59], energy expenditure [59], stress index [59], blood lactate [46], the Physiological Strain Index (PSI), Cumulative Heat Strain Index (CHSI), and exercise workload (measured as training impulse or TRIMP) [74].

HRV is the variation between heartbeats over time and is the close interplay between the sympathetic and parasympathetic nervous system within each heart rate cycle [94]. HRV was significantly associated with total sleep time and displayed an inverse relationship with sleepiness and fatigue [57]. Scores for HRV were suppressed in those on a 6-h rescue shift compared to those on a 24-h on-call shift and a 6-h ambulance-only shift [59]. Interestingly, heart rate and HRV showed no difference in on-call firefighters for those that had calls during the night compared to those that did not [65]. This trend continued when peak and average heart rates were found to have no observable difference between those in a wildfire deployment [74], between groups of firefighters in a sleep-restricted state or in a non-restricted sleep state during a simulated fire suppression [80,81,82], with or without a night call [65], or between trained and untrained firefighters [46]. However, when firefighters were on a 24-h “on-call” shift or on a 6-h shift of rescue calls, both mean and peak heart rates were increased [59]. Similarly, peak and mean VO_2Max_ was elevated in those on a 24-h on-call as it was more physiologically demanding than a 6-h shift of rescue or ambulance service [59]. In addition, when other objective physiological measurements were reported no change was seen in galvanic skin response [61], core body temperature [74,80,82], or blood lactate when compared to baseline shift levels [46].

#### 3.3.4. Cognitive Reaction Test

Nine studies reported on 13 cognitive reaction tests [49,52,56,57,62,63,66,67,75], the most common being the 5-min Psychomotor Vigilance Task (PVT) on a hand-held device which was reported in five studies [49,52,56,66,75]. Others included a simple, choice and discrimination reaction time test [57], Central Nervous System Vital Signs [63], the Sprawności Operacyjnej Test [62], D2 Test and the Colour Trails Test (CTT) [67], and the Go/No-go, the Stroop Colour Word Test, and the Occupational Safety Performance Assessment Test [49].

Studies found that those with sleep restriction showed poorer scores in the PVT [56,66,75], with performance decrements increasing as deployment length continued [56]. The PVT was the most sensitive objective test of fatigue when compared to the Go/No-Go, the Stroop Colour Word Test, and the Occupational Safety Performance Assessment Test [49]. Another sensitive measure to fatigue was the Sprawności Operacyjnej Test, as it identified more errors committed and fewer correct responses as total sleep deprivation increased from the baseline to 31-h awake [62]. Conversely, only one study [52] found no difference in PVT scores following three 24-h shift cycles when tested on the firefighters’ off day.

Regardless of shift type (3-, 6-, 9-, or 21-day cycle), visual memory, complex attention, composite memory, psychomotor speed, and motor speed significantly decreased the following day after a night shift using the Central Nervous System Vital Signs [63]. In contrast, the simple, choice, and discrimination reaction time found no associations with HRV being able to predict cognitive performance over a 14-day wildland firefighter deployment [57]. Similar findings were reported with the D2 and CTT, finding no differences between groups of firefighters, paramedics, or the control group of white-collar office workers [67].

#### 3.3.5. Self-Rated Measures

Self-rated measures of fatigue were conducted by nine studies using 11 different measures. Five different studies used the visual analogue scale (VAS) to assess fatigue [56,57], physical and cognitive fatigue [50], alertness [56,57], sleepiness [56,57], ability to predict performance [49,75], and sleep quality [56]. Another four studies used the rating of perceived exertion (RPE) [46,79,80,81] to rate physical task performance during a fatiguing task.

The VAS for fatigue and sleepiness increased as alertness decreased in line with deployment length [56,57], which was associated with a decrease in sleep quality in firefighters [56]. In the same study of wildland firefighters, the elevated levels of self-reported fatigue were not relieved by two days of rest [56]. Both physical and cognitive fatigue increased under average work conditions and rose to high levels under strenuous conditions [50]. Furthermore, when firefighters were asked to predict their cognitive performance, scores varied significantly [49,75]. Interestingly, better performing individuals were worse at predicting their performances than those with actual lower performance scores [75].

The RPE showed little to no difference when in a fully slept state or sleep-restricted state on physical task performance [80,81,82]. Conversely, the RPE and time to competition showed an increase after completing a simulated fire ground test compared to the baseline in both firefighters that exercise regularly and those that do not [46]. However, the trained firefighters performed the simulated fire ground test faster than 70% of the untrained firefighters [46].

#### 3.3.6. Physical Activity

Eleven measures of physical activity and its effects on fatigue were used in 10 studies [41,43,45,59,60,61,62,65,67,82]. Activity monitors reporting objective data were worn by participants in five studies [43,59,61,62,82], while six studies used subjective measures including the International Physical Activity Questionnaire-Short Form (IPAQ) [41], the Habitual Physical Activity Questionnaire (HPA-Q) [45], the General Health Questionnaire (SF-36) [60], Habitual Behaviour Inventory (HBI) [67], and task performance [80,82]. While the SF-36 is a questionnaire of general health, it was included in this section as general health and physical activity are intimately linked in EFRs [7].

During a single rotating shift roster, paramedics displayed a significant increase in step count on the first night shift compared to pre-shift [61]. However, no differences in total steps were found between those in a sleep-restricted state or non-sleep-restricted state [62]. Furthermore, when physical activity was measured with time to complete a task, in a sleep-restricted state or non-sleep-restricted state, no significant differences were found [80,81,82]. Similarly, performing physical work in high temperatures (33–35 °C) did not impact sleep beyond restricting sleep alone in firefighters [43]. Additionally, objectively assessing total energy expenditure was not statistically different during any shift compared to any time point of day or night shift [61].

Physical activity measured subjectively outside of work with the IPAQ and SF-36 found that paramedics were physically less active than the general public [41,60]. This trend was also observed by the HBI, which showed paramedics have an overall decrease in health practices compared to firefighters and office workers [67]. Conversely, 75% of military firefighters meet the recommended weekly amount of exercise by engaging in at least one type of moderate to vigorous exercise of at least 150 min, with 35% engaging in two physical exercises per week [45].

#### 3.3.7. Workload and Safety

Workload and safety measures were assessed by eight studies using 10 different measures [47,48,53,64,67,71,77,79] including the Emergency Medical Services Safety Inventory (EMS-SI) [47,48,64,71], workload and injury [64], perceived workload (physical and psychological) [67], perceived near-misses [77], and safety behaviour items from the National Fire Protection Agency 1500^TM^ (NFPA 1500^TM^)m along with perceived safety climate questions [53].

The EMS-SI showed that most respondents reported safety-compromising behaviours, all associated with fatigue and sleepiness [47,48,64,71]. In the EMS-SI, despite the heavy workload, there were no significant relationships between injury and workload when workload questions were assessed [64]. However, near-miss incidents were related to high levels of fatigue and posed an increased risk of occupational injury [77].

Safety concerns were found in two studies with the youngest workers reporting more adverse events [48,71], while another study [64] reported that those older and with more health concerns sustained more injuries. Interestingly, in petrochemical firefighters, how the firefighters perceived the workplace safety culture had an effect on safety behaviour [53]. Although, fatigue levels were low in petrochemical firefighters the authors state that improvement of fatigue can increase safety behaviour in the workplace [53].

Perceived workload is relative within the occupational realm. Firefighters, paramedics, and office workers all rated cognitive workload similarly [67]. However, both firefighters and paramedics rated higher levels of physical workload than office workers [67].

#### 3.3.8. Balance and Strength

Five measures for balance and strength were conducted in four studies [51,52,54,72], including the assessment of static and dynamic balance [51,72], the Y-balance Test [51], maximal isometric knee extension strength [54], and maximal rapid force production [52]. After completing a fatiguing protocol, the study by Games et al. [51] found that double-legged displacement, single leg sway, and anterior reach on the Y-Balance mean differences were 1.3 + 2.8 cm^2^, 2.3 + 4.5 cm^2^, and 1.5 + 2.6 cm, respectively, showing physical fatigue negatively impacted static and dynamic balance. Meanwhile, balance was further negatively impacted for on-call firefighters compared to salaried firefighters [72].

Regarding strength, no associations between maximal isometric knee extension and acute or chronic fatigue in firefighters were identified [54]. In contrast, Gerstner et al., [52] demonstrated that rapid strength, tested in less than 50 milliseconds (ms), was markedly decreased following a three 24-h shift cycle when tested on the firefighters’ off day. However, no differences were found in reactive strength at any timeframe after 50 ms [52].

### 3.4. Indirect Contributors

Two themes of the indirect contributors to fatigue were identified: mental health and recovery practices. Within the indirect effects of fatigue, there were 19 outcome measures assessed with one interview format based on eleven studies [41,45,48,50,59,60,61,63,66,73,76]

#### 3.4.1. Mental Health

Sixteen unique measures of various aspects of mental health were identified in eight studies [41,45,48,60,61,63,73,76]. Five studies assessed depressive symptoms [41,60,63,73,76], while the Beck Depression Inventory was reported twice [60,76], as was the Depression Anxiety Scale 21 (DASS21) [41,73]. Other measures included the Patient Health Questionnaire-9 (PHQ-9) [63], the Impact Event Scale (IES) [73], the Paschoal and Tamayo Work Stress Scale [45], the Positive and Negative Affect Scale (PANAS) [61], the Pittsburgh Sleep Quality Index-Addendum for Post-Traumatic Stress Disorder (PSQI-PTSD) [60], the Emergency Medical Services-Chronic Stress Questionnaire (EMS-CSQ), and the Post-Traumatic Stress Disorder (PTSD) checklist [48].

High rates of depression were found in 15% of the firefighters [63], while others reported it in a third of firefighters and paramedics [41,60,76]. Paramedics that subjectively rated increased fatigue displayed depressive symptoms [41,76], with depression being second only to sleep issues linked to fatigue [41,60]. Mood changes were not found to change across a shift schedule with the PANAS in a group of paramedics [61]; however, a rotating shift may be related to decreased sleep duration with increases in sleepiness, stress, and fatigue [61].

High-stress levels were a common occurrence in firefighters and paramedics [45,48], along with emotional trauma in both populations [48,60,73]. Stress and fatigue were significantly associated with injuries/exposures, safety-compromising behaviours, and errors/adverse events with a significant relationship to the organisational stress in paramedics [48]. Additionally, injuries and exposure to trauma were found to have a significant relationship in paramedics via the PTSD checklist [48]. PTSD, in paramedics, was reported at 16% and also considered to be a predictor of anxiety [60], with another quarter of paramedics reporting anxiety without PTSD [41].

#### 3.4.2. Recovery Practices

Two studies assessed fatigue recovery practices, through an open answer format [50] and using a recovery questionnaire [66]. A third study computed recovery scores based on objective data [59]. When firefighters were asked about recovery practices, the most-used recovery practices were sitting in the shade (93%), cold water ingestion (90%), and removing the helmet, flash hood, and jacket (89%) while on the fireground [50]. In the recovery questionnaire, self-reported recovery scores between deployment types in wildland firefighters were similar, with consistent scores regardless of the recovery opportunity time allocated each night [66]. Objective recovery scores were significantly lower for those only working a 6-h shift of rescue compared to those working a 6-h shift of ambulance calls or 24-h on-call emergencies [59].

## 4. Discussion

The aim of this review was to identify, synthesise, and critically appraise research on the main contributors to occupation-induced fatigue and its impacts on EFR. This was the first review identified that strove to analyse fatigue and performance within the EFR population. The findings within this review are strengthened by the overall low risk of bias of the selected studies. Generally, the studies included within this review were mostly based on cohort and cross-sectional studies with no randomised controlled trials included. The overall findings of studies reported in this review were graded Level III-2 based on the National Health and Medical Research Council (NHMRC) grading system.

Fatigue was prevalent throughout the majority of the studies included in this review and was reported both while on duty [41,42,44,45,47,48,49,50,56,57,61,66,68,70,71,73,76,77,78,79,83] and off duty [56,60]. Fatigue also led to increased feelings of depression and anxiety [41,60,76] and those that had a mental health concern reported higher levels of fatigue [41,69,73]. Additionally, those that were fatigued displayed a significant decrease in cognitive reaction speed [49,56,57,62,63,66,75]. Furthermore, fatigue was the most powerful influence in safety-compromising behaviours, injuries, and medication error or adverse events [47,48,64,71], and was identified as a contributor to reported near-miss incidents [77].

Overall, lack of sleep during the shift, poor sleep quality, and working through the night led to increased fatigue, decreased cognitive function, and decreased self-perceived alertness, while increasing safety-compromising behaviours for both the patient and the EFR. As sleep deprivation and fatigue increased, cognitive reaction slowed which caused a decrease in reaction time, situational awareness, and decision making [49,56,57,62,63,66,75]. This may be in part of the reason behind the aforementioned safety-compromising behaviours [47,48,64,71]. Furthermore, mental health issues plague EFRs [41,60,63,76] at a higher rate than in the general population [60]. Such issues are further compounded in EFRs who are regularly fatigued. Among the included studies in this review, there appears to be no consensus on the most applicable measures of fatigue, made evident by the 186 distinct outcome measures reported. Interestingly, no studies were identified reporting fatigue in emergency call centre personnel.

### 4.1. Outcome Measures for Fatigue Assessment

This review identified 186 outcome measures investigating fatigue in the included studies with most of the reported findings based on subjective outcome measures. Thus, the findings of this review strengthen the notion that subjective measures of fatigue using questionnaires such as the CFS, SPFS, and the OFERS (the three most commonly reported in this review) can aid in identifying occupation-induced fatigue. These findings align with evidence that the CFS can correctly identify those with mental and physical fatigue based on scores over the last month in healthcare workers [95]. When the CFS was measured repeatably over several months the scores were consistent in physicians [96] and nurses [97] making it a highly reliable assessment. Alternatively, the SPFS is a measure based on how the individual feels at the moment of assessment [32]. The SPFS is often utilised by the ICAO to inform performance ability with pilots on long-range and ultra-long-range flights to identify those at risk of fatigue while working [32]. Additionally, scores from the SPFS can detect accumulated fatigue with repeated testing on deployment in Navy personnel [98]. In contrast to the CFS and the SPFS, the OFERS assesses short-term (acute) and long-term (chronic) fatigue along with intershift recoverability [99]. The OFERS is recommended as a tool to identify those at risk of work-related stress and fatigue as it has the ability to quantify and distinguish between acute and chronic fatigue while simultaneously measuring recoverability [100]. Use of the aforementioned assessments would benefit EFR by identifying those at risk of immediate fatigue (SPFS), ongoing cognitive and physical fatigue (CFS), or short- and long-term fatigue with inadequate recovery (OFERS).

While subjective outcome measures are helpful, self-reports of workload and safety have their limitations as they have been shown to be less reliable once the individual is fatigued [32]. The self-rated measures used in this review, such as the VAS, were able to show that fatigue increased in line with the duration of work and sleep deprivation [50,56,57]; however, the ability to predict performance with self-reported assessments varied significantly [49,75]. Interestingly, better performing individuals were paradoxically worse at predicting their performances [75]. This may be in part due to optimism bias [101] whereby overestimating one’s ability helps one believe that there is control on future outcomes that can be forced into the direction desired. Once subjective fatigue has set in, the continued build-up of fatigue can exceed the individual’s capacity to adequately recover, leading to safety concerns [32,99]. The findings in this review align with the current literature on the ability to accurately self-perceive decrements in performance being challenging to assess [102,103,104]. The ability to make accurate self-ratings becomes increasingly unreliable even when performance and alertness decline [32].

To complement subjective measures, objective measures can be used to assess physical fatigue. In this review, heart rate was the most common objective assessment of physical fatigue, with no differences seen between firefighters during a wildfire suppression [74], sleep-restricted or non-restricted sleep state in a simulated fire suppression [80,81,82], between trained firefighters and untrained firefighters [46], or in an experimental design to induce fatigue in firefighters [65]. A potential reason for this lack of difference may lie in firefighters pacing themselves to match the physical demand needed by decreasing work performance when fatigued which would prevent heart rates from increasing [80]. Considering these findings, however, heart rate was found not to change during sleep even after having placed firefighters in a simulated fire intervention or real-life intervention during the middle of the night [65]. Based on these findings, the postulation by Rodríguez-Marroyo et al. [74], that the use of heart rate may not reflect the effort exerted due to the variable levels of the intensity of work done, bears merit. The findings of this review align with the current literature reported in a systematic review [105] of military personnel drawn from 20 studies with 2589 participants that stated heart rate monitoring is not yet conclusive for physical fatigue assessment. Heart rate has a multitude of influencing factors, including stress, body posture, and anxiety, and its wide range of variability shows that heart rate should not be used as a sole fatigue assessment [105].

Assessing physical fatigue can be difficult which lead some authors turning to assess cognitive fatigue. Assessing cognitive fatigue was most commonly completed with the 5-min PVT which consistently correlated to increasing levels of fatigue in firefighters [49,56,66,75]. The PVT provides insight into personal functioning by reporting instant feedback [75]. In addition, PVT scores continued to decline in line with wildland firefighters’ deployment length [56]. The findings of this review align with the current literature where cognitive fatigue reduces reaction time [16,106,107] and alertness [13]. Proper cognitive functioning can be negatively altered after just one night of sleep loss [32]. Even maintaining wakefulness for 17 h will reduce cognitive function by impairing reaction time [108]. Generally, the PVT can detect lapses in response time and measure the variability of reaction time which are indicators of the reduced ability to maintain alertness and attention [109].

### 4.2. Causes of Fatigue

While measuring fatigue is helpful, it is important to know the causes of fatigue. The key causes of fatigue identified in this review were lack of sleep on shift [56,57,58,61,66,67,79,82,83], poor sleep quality both on and off shift [41,45,55,58,60,65,67,68,71,76], and inadequate recovery between shifts [56]. Generally, poor sleep quality was identified predominantly through the PSQI [41,45,60,61,67,70,71,76,92], the most common outcome measure found in this review. Of note, the PSQI assesses sleep quality over the last 30 days. Besides poor sleep quality, authors reported other causes of fatigue were due to total sleep deprivation and the accumulation of a sleep debt [56,61,92]. A sleep debt accumulates when total sleep is consistently incomplete or reduced in quality [32,110]. Once a sleep debt has accumulated, the recovery of a normal sleep pattern may take at least two nights to dissolve [32]. Unfortunately, even with a large sleep debt, poor sleep quality as reported by many in this review [41,45,55,58,60,65,67,68,71,76], may be compromised due to insomnia [69].

Insomnia can be attributed to consistent sleep disturbances [111] and therefore causes an increase in fatigue levels [112]. Insomnia was reported in moderate levels by firefighters [45,63] and paramedics [60,69]. The insomnia reported was shown to cause a decrease in cognitive reaction speed and memory [63] and increases in depression and anxiety [60]. However, those that did not have insomnia based on a clinical diagnosis, yet maintained a poor sleep quality, were reported in greater numbers than those that did have a diagnosis of insomnia [60,63,68,76]. Decreased sleep quality both on and off shift was the reason for inadequate recovery between shifts [56]. The findings of this review align with the current literature that consistent sleep deprivation, sleep loss, or decreased sleep quality causes increased cognitive and physical fatigue in firefighters [113] and paramedics [5,17].

### 4.3. Impacts of Fatigue

Several emerging impacts of fatigue were identified in this review. These were reduced physical activity [41,60], balance [51,72], rapid (<50 ms) force production [52], cognitive performance [49,56,57,62,63,66,75], and increased emotional dysfunction [41,60,73,76], daytime sleepiness [64,76], sleepiness on off days [56,61,76], and negative safety impacts [47,48,64,71]. The requirements of the job leave EFRs tired, making them less physically active than the general public [41,60] which raises concern, as the health benefits of physical activity are well-documented [114]. Additionally, paramedics displayed higher overall activity during the night shift than the day shift which may put them at risk of physical fatigue [61].

One measure of physical fatigue was a reduction in firefighter balance [51,72]. Of note to this population, when fatigue was combined with heat, functional balance was further negatively affected [51]. Fatigue-related issues were further compounded for on-call firefighters placing them at an increased risk of balance-related injuries compared to their salaried counterparts [72]. Findings from this review also reported reductions in rapid (<50 ms) force production by 25% [52] during knee extension testing. Conversely, fatigue was found to not impact maximal force output generated in firefighters when testing knee extension strength [52,54]. Thus, force production rates (or power) decreased without a notable change in strength. Thus, with fatigue commonly accompanied by a decrease in rapid force production, outcome measures of rapid force production (e.g., vertical jump) have been suggested as valid indicators of physical fatigue in a recent scoping review [115] and in the wider literature [116]. Interestingly, sleep restriction did not appear to alter the physical task performance in firefighters during a simulation of less than 5-min bouts [81,82]. Further research will be needed to identify if longer than 5-min bouts will produce similar results. Generally, reduced balance and decreased force production may contribute to the high number of slips, trips, and fall injuries, which are reportedly the most common ways firefighters are injured while on duty [19] and account for 10–20% of all paramedic injuries [20].

The dangers of physical fatigue are important, but cognitive fatigue can strongly impact occupational performance as well [117]. Several studies in this review found that high levels of fatigue impacted cognitive performance negatively, such as decreased reaction times, and that cognitive function continued to decrease in line with sleep deprivation [49,56,57,62,63,66,75]. These findings align with current research that cognitive fatigue slows attentional awareness and increases accidents, performance errors, and poor decision making [118]. Similar results were found in shift workers [13,107,119,120,121], military personnel [31,122,123], the police [119], and athletes [14,124,125,126] demonstrating the negative impacts in cognitive performance due to fatigue.

Physical and cognitive fatigue notwithstanding, those who were fatigued were more likely to be negatively impacted with depression and anxiety [60]. Additionally, those with a mental health concern were more likely to be fatigued [41,69,73] creating a further downward spiral. Several authors in this review reported high numbers of depression, anxiety, and emotional trauma [41,60,73,76] at levels which exceed the general population [127]. Unfortunately, mental health concerns and fatigue may be further impacted by insomnia [60]. Findings of this review align with two systematic reviews that state EFRs with sleep issues are more likely to be impacted by depression and anxiety [2,128]. As previously discussed, insomnia causes poor sleep quality and poor sleep quality increases daytime sleepiness [60,64,70,76] and excessive sleepiness levels [64,70,76], which continue to impact EFRs on days off [56,61]. In addition, chronic sleep deprivation can increase negative emotions and negatively impact workplace efficiency and productivity [129].

Furthermore, daytime sleepiness impacted safety with reports of falling asleep at the wheel [73,76,79]. Fatigue and sleepiness negatively impacted safety [47,48,64,71] for paramedics [47,48,64,71] and firefighters [64] through compromised behaviours [47,48,71], medication errors [48,71], and increased injuries [47,48,64,71]. A staggering 90–96% of EFRs reported safety-compromised behaviours in their last three month period when surveyed [47,48,71], with perceived patient safety 4.9 times more likely to be negatively impacted when fatigued [71]. Furthermore, 50–76% of paramedics reported an adverse event or medication error in the last three months [47,48,71]. Adverse events or medication errors were 2.3 times more likely when fatigued vs. non-fatigued [71]. Additionally, injuries were 2.9 times more likely for those fatigued vs. non-fatigued [71]. Of the EFRs sampled, 33% [47] and up to 81% [48,64] reported having an injury in the last three months. The findings of this review align with the current research noting that paramedics can suffer 29 to 345 injuries per year per 1000 workers [20] and is considered to be one of the most dangerous professions in Australia [27]. Meanwhile in the US, career firefighters have an average of 69 injuries per 1000 workers per year, and 36% of all injuries are related to patient care [130].

### 4.4. Fatigue Mitigation Strategies

Due to the continuous demands of the operational nature of EFRs, fatigue is an inevitable part of the job and cannot be completely eliminated [53], but it is imperative that it is managed [32]. Preventing short- and long-term issues for fatigue risk management should include both personal and workplace socio-cultural risk factor assessments [7]. Comprehensive fatigue risk management strategies should include self-rated measures [75,103]; however, they should be in tandem with objective measures [32,103]. The combination of subjective and objective measures would allow high-risk personnel to be swiftly identified within the organisation. Early identification, coupled with a simple and straightforward process for reporting fatigue, and enforcing breaks to promote safety behaviour [53,79,83] may decrease adverse safety events. Four fatigue risk-mitigating strategies (shift cycles, cold water immersion, sleep hygiene, and exercise) are discussed below.

#### 4.4.1. Shift Cycles

Measuring fatigue may not be enough to stop fatigue as this only identifies when someone is already fatigued. In that case, mitigation strategies have been suggested in a recent study on guiding principles for shift work duration [12]. The authors recommended that risk mitigation could be maximised by aligning work schedules with circadian rhythms, protecting sleep opportunities, and/or increasing recovery time after multiple shifts or extended duties [12]. In nurses, those that worked a backward rotation displayed higher levels of fatigue and decreased cognitive performance compared to those working a forward rotation [131]. Similarly, in a systematic review of shift workers, a forward rotating schedule was superior at following the circadian rhythm at little to no cost from the organisation [132].

In this review, when workload demand was high with wildland firefighters on a 14-day deployment, 2 days off was not enough to alleviate fatigue levels [56]. The authors of the included studies in this review reported suggestions on ways to arrange shift cycles to avoid excess fatigue such as avoiding early start times [66], minimising scheduling that only allows for 8–9 h between shifts [83], or having less than 12 h between shifts [66]. Avoiding early start times is a strategy that has already been suggested in shift workers [121], pilots [133], firefighters, paramedics, and police [134].

#### 4.4.2. Sleep Hygiene

Inducing sleep can be difficult for a multitude of reasons. Sleep hygiene is an often-underutilised tool in education and self-awareness training that could be useful for those on lengthy suppression activities to become aware of fatigue symptoms to aid in sleep [66]. Sleep hygiene education has no negative documented effects [12] and includes awareness of behavioural and environmental contributors to increase sleep quality. A recent systematic review of 16 studies based on sleep hygiene and shift workers found that minimal attention has been paid to sleep hygiene [135]. Of the studies that did investigate sleep hygiene, there was minimal attention paid to the factors necessary for success as outlined by the Australian Sleep Association [135], which include daytime naps, regular exercise, eating a balanced diet, and avoiding TV before bed, alcohol 4 h before bed, caffeine 6 h before bed, and nicotine altogether.

A randomised control trial focused on sleep education training in EFR (n = 435) found a decrease in fatigue and an increase in sleep quality over a 3-month follow-up period [136]. Over the course of 3 months, various modules were completed on topics such as sleep physiology, sleep health, work-related stress, sleep disorders, hazards of fatigue, fatigue recognition, adequate sleep, diet and exercise, alertness strategies, and managing fatigue. The more modules the EFR watched the greater increase in sleep quality and reductions in fatigue were reported, albeit statistically small (Cohen’s d = −0.17, *p* = 0.05) [136]. However, this new information could help EFR agencies when designing fatigue risk management strategies to help increase sleep quality and reduce fatigue in this extremely vulnerable population [136].

#### 4.4.3. Cold Water Immersion

Recovery from physical fatigue during and between shifts can be difficult. One suggested strategy is cold water immersion (CWI). In both rugby and soccer players after playing a full game, CWI for ~10 min in 10 °C was shown to decrease physical fatigue as measured by an increase in rapid force production and peak power output [137,138]. CWI results were similar when measured subjectively in a systematic review of 99 papers, covering 1188 participants of various sports and backgrounds on ways to reduce perceived fatigue [139]. CWI may not only help enhance physical performance and reduce perceived fatigue, in the military, it may help reduce anxiety [140] with anxiety being a common finding associated with fatigue in this review. While outside of the scope of this review, positive mental health is essential in EFRs for overall job safety and quality of life [141,142]. Unfortunately, some barriers to cold-water immersion are that many organisations do not have baths available due to cost or being unaware of the methods [143]. In these cases, a cold shower is a cheap and practical alternative.

#### 4.4.4. Exercise

Findings included in this review suggested that maintaining higher levels of physical fitness by training on-duty may offset occupation-related physical fatigue stemming from poor fitness levels [46] as exercise helps avoid overload [59]. Within this review, a group of firefighters who exercised regularly had faster completion times in a simulated fire ground test compared to those that do not exercise even when fatigued [46]. The findings highlight the importance of regular on-duty exercise benefiting long-term occupational requirements of firefighters and may outweigh the short-term decrease in physical ability [46].

In pilots, those who engage in less physical activity are more likely to report increased fatigue levels [144]. In anaesthesiologists working the night shift, those that exercised regularly for 30–60 min every day were more likely to report lower physical fatigue with an increase in mental health status [145]. Similarly, nurses engaging in regular exercise reported lower levels of fatigue and an increase in mental health status [146]. Mental health issues (such as depression and anxiety [147,148]), insomnia [149], and sleep quality [122], which were found to increase fatigue in this review, have been found to be improved through regular low-intensity exercise.

However, increasing physical activity at the expense of sleep may worsen fatigue levels [150] or physical exhaustion [151]. Shift workers have stated barriers to exercise as lack of opportunity, time, and already being fatigued [150]. In paramedics, other barriers to exercise have been reported as a lack of willpower or energy [152]. Generally, those with increased levels of physical activity have been found to have an increase in occupational performance due to a reduced effort exerted to work [153]. Therefore, where possible, it is strongly recommended that EFR be given the opportunity to exercise while on shift to combat occupation-induced fatigue and the negative physical, cognitive, and mental health impacts that ensue.

### 4.5. Strengths and Limitations

A significant strength of this review is the large number of quality studies from around the world which makes the application of the findings more applicable to EFR generally. There was a considerable variation in the measures of fatigue which can appear to limit the conclusions in regards to the applicability of fatigue measures but strengthens the findings of the current literature that there are no clear best measures of fatigue. There are several limitations with this review: (1) most objective data is based on wildland firefighters that did not work a night shift; (2) high-quality studies with mixed populations were excluded; (3) many of the studies that used subjective outcome measures recounted past events which can introduce recall bias [154]; (4) due to heterogeneity of the data a meta-analysis was unable to be completed; (5) while wearing clothing and equipment will increase fatigue levels [50], studies that were only exploring how firefighter-specific clothing and equipment impact fatigue were excluded; (6) no cognitive reaction testing was conducted on paramedics; (7) only articles in English (or translatable to English) were included which can introduce language bias. However, given the number of articles informing this review this lack would be postulated to not impact conclusions [155]; (8) only one researcher conducted the search, selection of studies, and extracted the data, which may increase selection bias [155].

### 4.6. Implications for Practice and Policy

Findings from this review emphasise the current literature that both subjective and objective measures should be utilised to assess fatigue. Subjective measures should include physical and cognitive fatigue along with the objective measures of physical and cognitive fatigue. This would ensure a holistic assessment with enough overlap to identify those at risk of fatigue for early risk mitigation. Although the inclusion of such measures can be costly, the initial investment into the early identification of fatigue may reduce injury risk, which may save departments more money than the initial investment costs. Although cognitive testing is essential to fatigue management, most assessments do not test decision making or situational awareness. Finally, any single methodology used to assess fatigue has limitations that need to be recognised due to the complexity of fatigue.

### 4.7. Future Research

This review has demonstrated the need for more objective sleep monitoring during shift work (TST), cognitive assessments (PVT), and subjective fatigue questionnaires (i.e., CFS, SPFS, OFERS) to be implemented for early fatigue recognition while on-duty. Within this review, only two studies assessed physical fatigue for the lower body while no studies measured physical fatigue in the upper limb, such as grip strength, which should be explored as both paramedics and firefighters have a high demand usage of the upper limbs. Future studies should focus on firefighters and paramedics as this population is far less explored than tactical operators (police and military). Similarly, there were no paramedic studies that looked at cognitive reaction time which is an area for future research. Emergency call centre staff is another population that should be focused on as no studies were identified through this systematic search. This population has no physicality within their job but may have increased cognitive, visual, and emotional fatigue. In addition, sleep hygiene education and fatigue management strategy educational sessions for both EFR and managerial staff should be assessed for usefulness. Future research should include cognitive and physical fatigue measures to identify and discover the best outcome measures to evaluate performance capabilities as there is no clear consensus for the best fatigue measurements. Additionally, future firefighter-specific fatigue research should investigate how gear affects fatigue and its contribution to safety risks. Finally, more studies need to report on fatigue as this will largely impact occupational performance.

## 5. Conclusions

The main contributors to occupation-induced fatigue were lack of sleep during the shift and consistent poor sleep quality, which negatively impacted cognitive function, alertness, and physical and mental health while increasing safety-compromising behaviours and injuries. As sleep deprivation leads to fatigue, sleep was most commonly measured subjectively with the Pittsburgh Sleep Quality Index and objectively with wearable sleep monitors. In addition, fatigue was most commonly measured subjectively with the Chalder Fatigue Scale, while objectively being measured with the Psychomotor Vigilance Task and heart rate for cognitive and physical fatigue, respectively. Furthermore, there was no clear consensus on the most applicable measures or a single measurement to assess fatigue. Therefore, subjective and objective measures should be used in tandem as part of a comprehensive fatigue risk management plan.

In conjunction with having subjective and objective measures, implementing organisational fatigue-reducing strategies may help mitigate the negative impacts. It is therefore recommended that EFRs be encouraged to exercise while on shift to help increase sleep quality, physical and cognitive health, and reduce fatigue. The benefits will allow EFRs to continue to protect the public they serve while minimising risky outcomes.

## Figures and Tables

**Figure 1 ijerph-20-07055-f001:**
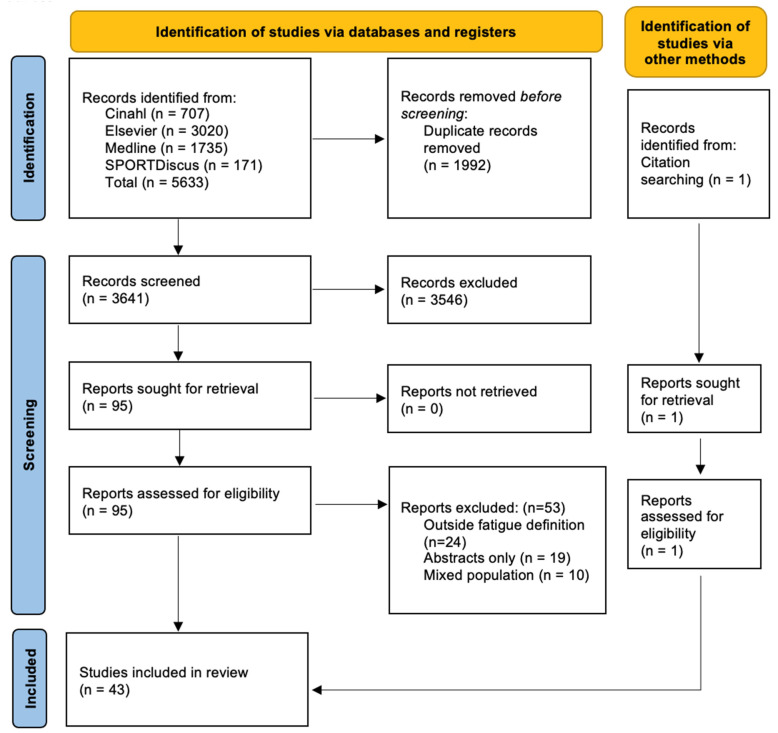
The Preferred Reporting Items for Systematic Reviews and Meta-Analyses (PRISMA) flowchart.

**Table 1 ijerph-20-07055-t001:** Search terms.

Database	Search Terms
Medline	(((“exertion” [Title/Abstract] OR “no rest” [Title/Abstract] OR “exhaustion” [Title/Abstract] OR “strain” [Title/Abstract] OR “muscle fatigue” [Title/Abstract] OR “Sleep deprivation” [Title/Abstract] OR “rest” [Title/Abstract] OR “wakefulness” [Title/Abstract] OR “fatigue*” [Title/Abstract] OR “muscle fatigue” [MESH] OR “Sleep Deprivation” [MESH] OR “Rest” [Mesh] OR “Wakefulness”[Mesh] OR “fatigue” [Mesh])) AND ((“Fire*” [Title/Abstract] OR “paramed*” [Title/Abstract] OR “call centers” [Title/Abstract] OR “Ambulance” [Title/Abstract] OR “first responder” [Title/Abstract] OR “EMS” [Title/Abstract] OR “emergency medical technicians” [Title/Abstract] OR “emergency medical dispatch” [Title/Abstract] OR “Emergency Medical Technicians” [Mesh] OR “Firefighters” [Mesh] OR “Call Centers” [Mesh] OR “Emergency Medical Dispatch” [Mesh] OR “Ambulances” [Mesh]))) NOT ((“Animals” [Mesh] NOT (“Animals” [Mesh] AND “Humans” [Mesh])))
Embase	(exertion:ti,ab OR ‘no rest’:ti,ab OR exhaustion:ti,ab OR strain:ti,ab OR ‘muscle fatigue’:ti,ab OR ‘sleep deprivation’:ti,ab OR rest:ti,ab OR wakefulness:ti,ab OR fatigue*:ti,ab OR ‘muscle fatigue’/exp OR ‘sleep deprivation’/exp OR ‘rest’/exp OR ‘wakefulness’/exp OR ‘fatigue’/exp) AND (fire*:ti,ab OR paramed*:ti,ab OR ‘call centers’:ti,ab OR ambulance:ti,ab OR ‘first responder’:ti,ab OR ems:ti,ab OR ‘emergency medical technicians’:ti,ab OR ‘emergency medical dispatch’:ti,ab OR ‘rescue personnel’/exp OR ‘fire fighter’/exp OR ‘call center’/exp OR ‘emergency medical dispatch’/exp OR ‘ambulance’/exp) NOT ([animals]/lim NOT [humans]/lim)
CINAHL	((TI exertion OR AB exertion) OR (TI “no rest” OR AB “no rest”) OR (TI exhaustion OR AB exhaustion) OR (TI strain OR AB strain) OR (TI “muscle fatigue” OR AB “muscle fatigue”) OR (TI “Sleep deprivation” OR AB “Sleep deprivation”) OR (TI rest OR AB rest) OR (TI wakefulness OR AB wakefulness) OR (TI fatigue* OR AB fatigue*) OR (MH “muscle fatigue”+) OR (MH “Sleep Deprivation”+) OR (MH Rest+) OR (MH Wakefulness+) OR (MH fatigue+)) AND ((TI Fire* OR AB Fire*) OR (TI paramed* OR AB paramed*) OR (TI “call centers” OR AB “call centers”) OR (TI Ambulance OR AB Ambulance) OR (TI “first responder” OR AB “first responder”) OR (TI EMS OR AB EMS) OR (TI “emergency medical technicians” OR AB “emergency medical technicians”) OR (TI “emergency medical dispatch” OR AB “emergency medical dispatch”) OR (MH “Emergency Medical Technicians”+) OR (MH Firefighters+) OR (MH “Emergency Medical service communications”+) OR (MH Ambulances+))
SPORTDiscus	((TI “exertion” OR AB “exertion”) OR (TI “no rest” OR AB “no rest”) OR (TI “exhaustion” OR AB “exhaustion”) OR (TI “strain” OR AB “strain”) OR (TI “muscle fatigue” OR AB “muscle fatigue”) OR (TI “Sleep deprivation” OR AB “Sleep deprivation”) OR (TI “rest” OR AB “rest”) OR (TI “wakefulness” OR AB “wakefulness”) OR (TI “fatigue*” OR AB “fatigue*”) OR DE “Sleep Deprivation” OR DE “Rest” OR DE “fatigue”) AND ((TI “Fire*” OR AB “Fire*”) OR (TI “paramed*” OR AB “paramed*”) OR (TI “call centers” OR AB “call centers”) OR (TI “Ambulance” OR AB “Ambulance”) OR (TI “first responder” OR AB “first responder”) OR (TI “EMS” OR AB “EMS”) OR (TI “emergency medical technicians” OR AB “emergency medical technicians”) OR (TI “emergency medical dispatch” OR AB “emergency medical dispatch”) OR DE “Emergency Medical Technicians” OR DE “Call Centers” OR DE “Emergency Medical Dispatch” OR DE “Ambulances”)

**Table 2 ijerph-20-07055-t002:** Demographics of included studies.

Author/Year	Participants	Study Design
Courtney et al., 2013 [41]	Paramedics; n = 148, m = 117, f = 31	Cross-sectional
Courtney et al., 2020 [42]	Firefighters; n = 39, m = 34, f = 4, unknown sex = 1, 38.75 ± 10.60 yr, 190.65 ± 28.51 lb, 27.53 ± 3.38 kg/m^2^	Cohort
Cvirn et al., 2017 [43]	Volunteer firefighters; n = 61, m = 52, f = 9 Control; n = 25, m = 22, f = 3, 36 ± 15.9 yr, 27 ± 4.8 kg/m^2^ Awake; n = 25, m = 20, f = 5, 38.5 ± 13.2 yr, 29.2 ± 4.9 kg/m^2^ Awake/hot; n = 11, m = 10, f = 1, 37.5 ± 15.6 yr, 26.7 ± 4.6 kg/m^2^	Cohort
Dawson et a., 2015 [44]	Volunteer firefighters; n = 30, unknown sex = 30, 21–65 yr	Qualitative
de Carvalho Dutra 2017 [45]	Military firefighters; n = 20, m = 16, f = 4, 26–45 yr, 28.2 kg/m^2^	Cross-sectional
Dennison et al., 2012 [46]	Firefighters; n = 49, m = 49, Trained; 31.8 ± 6.9 yr, 87.6 ± 14.3 kg, 27.7 ± 3.3 kg/m^2^ Untrained; 31 ± 9 yr, 102 ± 19.5 kg, 31.3 ± 5.2 kg/m^2^	Cohort
Donnelly et al., 2019 [47]	Paramedics; n = 717, m = 474, f = 243	Cross-sectional
Donnelly et al., 2020 [48]	Paramedics; n = 717, m = 474, f = 243, 38 ± 10.1 yr	Cross-sectional
Ferguson et al., 2016 [49]	Firefighters/volunteers; n = 88, m = 77, f = 11, 38.4 ± 14.4 yr, 27.8 ± 4.53 kg/m^2^	Cohort
Fullagar et al., 2021 [50]	Firefighters; n = 473, m = 417, f = 51, unknown sex = 5, 46 ± 11 yr	Cross-sectional
Games et al., 2020 [51]	Firefighters; n = 41, m = 41, 37 ± 8.1 yr, 98.8 ± 14.3 kg	Cohort
Gerstner et al., 2022 [52]	Firefighters; n = 35, m= 32, f = 3, 34 ± 9 yr, 97 ± 21 kg, 30 ± 5.5 kg/m^2^	Cohort
Ghasemi et al., 2021 [53]	Petrochemical firefighters; n = 261, m = 261, 36.68 ± 6.80 yr	Cross-sectional
Giuliani et al., 2020 [54]	Firefighters; n = 32, m = 29, f = 3, 33.7 ± 9.2, 94.5 ± 20.8 kg, 30 ± 5.5 kg/m^2^	Cohort
Huang et al., 2022 [93]	Firefighters; n = 801, m = 801, 32 ± 7.2 yr, 25 ± 3.7 kg/m^2^	Cross-sectional
Jeklin et al., 2020 [56]	Firefighters; n = 30, m = 20, f = 10, 24.6 ± 4.8 yr, 25.9 ± 3.2 kg/m^2^	Cohort
Jeklin, Davies, et al., 2021 [83]	Wildland firefighters; n = 39, m = 26, f = 13, 30.4 ± 11.6 yr	Cohort
Jeklin, Perrotta, et al., 2021 [57]	Wildfire services; n = 9, m = 6, f = 3, 48.5 ± 6.4 yr, 84.1 ± 19.0 kg, 28.9 ± 5.3 kg/m^2^	Cohort
Jeong et al., 2019 [58]	Firefighters; n = 294, m = 269, f = 25, <40 yr =153, 40–49 yr = 94, >50 yr = 47, BMI < 25 = 196 and >25 = 98	Cross-sectional
Kaikkonen et al., 2017 [59]	Firefighters; n = 21, unknown sex = 21, 38 ± 7 yr, 79 ± 10 kg, 25 ± 2 kg/m^2^	Cohort
Khan et al., 2020 [60]	Paramedics; n = 134, m = 72, f = 62, 39.1 ± 12.1 yr, 26.7 ± 4.9 kg/m^2^	Cross-sectional
Khan et al., 2021 [61]	Paramedics; n = 12, m = 5, f = 7, 39.5 ± 10.7 yr, 24.5 ± 3.4 kg/m^2^	Cohort
Kujawski et al., 2018 [62]	Firefighters; n = 55, m = 55, 32.6 ± 6.8, 24.6 ± 2.6 kg/m^2^	Cohort
Kwak et al., 2020 [63]	Firefighters; n = 352, m = 328, f = 24, 40.1 ± 8.7 yr	Cohort
Lin et al., 2020 [64]	EMT; n = 347, m = 334, f = 13, 20–29 yr, BMI 18.5–24 = 135, BMI > 24 = 212	Cross-sectional
Marcel-Millet et al., 2020 [65]	Firefighters; n = 13, m = 13, 36.3 ± 6.2 yr, 73.7 ± 9.4 kg, 23.9 ± 1.7 kg/m^2^	Cohort
McGillis et al., 2017 [66]	Wildland firefighters; n = 21, m = 21, 29.9 ± 8.4 yr	Cohort
Nowak and Łukomska, 2021 [67]	Paramedics; n = 18, m = 12, f = 6, 31.83 ± 4.73 yr Firefighters; n = 15, m = 15, 33 ± 5.61 yr	Cohort
Paterson et al., 2016 [68]	Firefighters; n = 46, m = 46, Salaried; 38 ± 10 yr Retained; 33 ± 8 yr	Qualitative
Paterson et al., 2014 [69]	Paramedics; n = 49, m = 37, f = 12, 38 ± 9.7 yr	Cross-sectional
Patterson et al., 2016 [70]	Paramedics; n = 1, m = 1, 26 yr, 29.5 kg/m^2^	Case-report
Patterson et al., 2012 [71]	EMT; n = 511, m = 378, f = 133, 37 ±10.6 yr, BMI 18.5–24.9 = 112, BMI 25–30+ = 396	Cross-sectional
Pau et al., 2014 [72]	Firefighters; n = 26, m = 26 Career; 46.2 ± 4.7 yr, 26.3 ± 2.7 kg/m^2^ Retained; 29.1 ± 6.1 yr, 26.6 ± 3.4 kg/m^2^	Cohort
Pyper and Paterson, 2016 [73]	Paramedics; n = 134, m = 103, f = 31, 21–60+ yr	Cross-sectional
Rodríguez-Marroyo et al., 2012 [74]	Firefighters; n = 160, unknown sex = 160, 25.2 ± 0.4 yr, 75.8 ± 0.8 kg, 24.3 ± 0.2 kg/m^2^	Cohort
Smith et al., 2016 [75]	Rural firefighters; n= 91, m = 79, f = 12, 38.4 ± 14.4 yr, 27.8 ± 4.53 kg/m^2^	Cohort
Sofianopoulos et al., 2011 [76]	Paramedics; n = 60, m = 46, f = 14, 21–45+ yr	Cross-sectional
Toyokuni et al., 2022 [77]	Paramedics; n = 254, m = 254, 18–50+ yr	Cross-sectional
Vincent et al., 2016 [78]	Wildland firefighters; n = 33, m = 25, f = 8	Cohort
Vincent et al., 2016 [79]	Wildfire firefighters; n = 40, m = 31, f = 9, 11.1 ± 11, 26.8 ± 4.7 kg/m^2^	Cohort
Vincent et al., 2017 [80]	Wildland firefighters; n = 30, m = 27, f = 3, Sleep restricted; n = 17, 93.8 ± 20.2 kg, 29.6 ± 5.5 kg/m^2^ Hot and sleep restricted; n = 13, 83.8 ± 14.3 kg, 27.0 ± 4.3 kg/m^2^	Cohort
Vincent et al., 2018 [81]	Firefighters/volunteers; n = 31, m = 26, f = 5 Hot condition; n = 18, 36 ± 13 yr, 88.0 ± 18.0 kg, 27.5 ± 3.5 kg/m^2^ Hot + sleep restricted; n = 13, 41 ± 17 yr, 83.8 ± 14.3 kg, 27.0 ± 4.3 kg/m^2^	Cohort
Vincent et al., 2015 [82]	Firefighters; n = 35, m = 30, f = 5 Control; 39 ± 16 yr, 85.1 ± 7.7 kg, 26.7 ± 4.8 kg/m^2^ Sleep restricted; 39 ± 15 yr, 93.8 ± 20.2 kg, 29.6 ± 5.5 kg/m^2^	Cohort

n = total number of participants; m = male; f = female; kg = kilograms of body weight; yr = years; kg/m^2^ = kilograms per metre squared for body mass index.

**Table 3 ijerph-20-07055-t003:** Joanna Briggs Institute scores: Cohort studies.

Authors	Questions																			Overall Score
	1		2	3	4		5	6		7		8		9		10		11		
Courtney et al., 2020 [42]	N/A		+	+	+		-	-		+		+		+		+		+		80%
Cvirn et al., 2017 [43]	+		+	+	+		-	+		+		+		+		-		+		82%
Dennison et al., 2012 [46]	+		+	+	+		-	+		+		+		+		N/A		+		90%
Ferguson et al., 2016 [49]	+		+	+	-		-	+		+		+		+		-		+		73%
Games et al., 2020 [51]	N/A		+	+	+		-	+		+		+		+		+		+		90%
Gerstner et al., 2022 [52]	N/A		+	+	+		-	+		+		+		+		+		+		90%
Giuliani et al., 2020 [54]	N/A		+	+	+		+	-		+		+		+		+		+		90%
Jeklin et al., 2020 [56]	N/A		+	+	+		-	+		+		+		+		+		+		90%
Jeklin, Davies, et al., 2021 [83]	N/A		+	+	-		-	+		+		+		+		+		+		80%
Jeklin, Perrotta, et al., 2021 [57]	N/A		+	+	+		-	-		+		+		+		-		+		70%
Kaikkonen et al., 2017 [59]	N/A		+	+	+		-	-		+		+		+		-		+		70%
Khan et al., 2021 [61]	N/A		+	+	-		-	+		+		+		+		+		+		80%
Kwak et al., 2020 [63]	+		+	+	+		+	+		+		+		+		+		+		100%
Kujawski et al., 2018 [62]	N/A		+	+	-		-	+		+		+		+		-		+		70%
Marcel-Millet et al., 2020 [65]	N/A		+	+	+		-	+		+		+		-		-		+		70%
McGillis et al., 2017 [66]	N/A		+	+	+		-	+		+		+		+		-		+		80%
Nowak and Łukomska, 2021 [67]	+		+	+	+		-	-		+		+		+		+		+		82%
Pau et al., 2014 [72]	+		+	+	+		+	+		+		+		+		+		+		100%
Rodríguez-Marroyo et al., 2012 [74]	N/A		+	+	-		-	+		+		+		-		-		+		60%
Smith et al., 2016 [81]	+		+	+	+		-	+		+		+		-		-		+		73%
Vincent et al., 2015 [82]	+		+	+	+		+	+		+		+		+		+		+		100%
Vincent et al., 2016 [78]	N/A		+	+	+		-	-		+		+		+		-		+		70%
Vincent et al., 2016 [79]	N/A		+	+	-		-	+		+		-		+		+		+		70%
Vincent et al., 2017 [80]	+		+	+	+		+	+		+		+		+		-		+		90%
Vincent et al., 2018 [57]	+		+	+	+		-	+		+		+		-		-		+		73%
*Cross-sectional studies*																				
**Author**	**Questions**																		**Overall score**	
	**1**	**2**		**3**			**4**		**5**		**6**				**7**		**8**			
Courtney et al., 2013 [41]	+	-		+			+		-		-				+		+		63%	
de Carvalho Dutra, 2017 [45]	+	+		+			+		-		-				+		+		75%	
Donnelly et al., 2019 [47]	+	-		+			+		+		+				+		-		75%	
Donnelly et al., 2020 [48]	-	+		+			+		+		+				+		+		88%	
Fullagar et al., 2021 [50]	+	+		-			+		+		-				-		+		63%	
Ghasemi et al., 2021 [53]	-	+		+			-		+		+				+		+		75%	
Huang et al., 2022 [55]	+	+		+			+		+		+				+		+		100%	
Jeong et al., 2019 [78]	+	+		+			+		+		-				+		+		88%	
Khan et al., 2020 [43]	+	+		+			+		+		+				+		+		100%	
Lin et al., 2020 [64]	+	+		+			+		+		+				+		+		100%	
Paterson et al., 2014 [69]	-	+		+			-		+		-				+		+		62%	
Patterson et al., 2012 [71]	-	+		+			+		+		+				+		+		88%	
Pyper and Paterson, 2016 [73]	-	+		+			-		-		-				+		-		37%	
Sofianopoulos et al., 2011 [76]	+	+		+			+		-		-				+		+		75%	
Toyokuni et al., 2022 [77]	+	+		-			-		+		-				-		+		50%	
*Qualitative studies*																				
**Author**	**Questions**																		**Overall score**	
	**1**		**2**	**3**		**4**	**5**		**6**		**7**		**8**		**9**		**10**			
Dawson et al., 2015 [44]	+		+	+		+	+		-		-		+		+		+		90%	
Paterson et al., 2016 [68]	+		+	+		+	+		-		+		+		+		+		90%	
*Case-report study*																				
**Author**	**Questions**																		**Overall score**	
	**1**	**2**		**3**			**4**		**5**		**6**				**7**		**8**			
Patterson et al., 2016 [70]	+	+		+			+		+		+				-		+		88%	

N/A = not applicable, ‘+’ = yes, ‘-‘ = no. The full questions for the Joanna Briggs Institute cohort studies checklist can be found at https://jbi.global/sites/default/files/2019-05/JBI_Critical_Appraisal-Checklist_for_Cohort_Studies2017_0.pdf accessed on 6 November 2023; The full questions for the Joanna Briggs Institute cross-sectional studies checklist can be found at https://jbi.global/sites/default/files/2019-05/JBI_Critical_Appraisal-Checklist_for_Analytical_Cross_Sectional_Studies2017_0.pdf accessed on 6 November 2023; The full questions for the Joanna Briggs Institute qualitative studies checklist can be found at https://jbi.global/sites/default/files/2019-05/JBI_Critical_Appraisal-Checklist_for_Qualitative_Research2017_0.pdf accessed on 6 November 2023; The full questions for the Joanna Briggs Institute case-report study checklist can be found at https://jbi.global/sites/default/files/2019-05/JBI_Critical_Appraisal-Checklist_for_Case_Reports2017_0.pdf, accessed on 6 November 2023.

**Table 4 ijerph-20-07055-t004:** Outcomes of fatigue.

Author	Fatiguing Variable	Acute Fatigue, Chronic Fatigue or Combined	Outcome Measures	Results	Summary
Cvirn et al., 2017 [43]	3-day 4-night experiment conditions: (1) 8-h sleep control (2) 4-h sleep (3) 4-h sleep + heat	Acute fatigue	Activity monitor Polysomnography (PSG): total sleep time (TST), sleep onset latency (SOL), wake after sleep onset (WASO), light sleep (1 and 2), deep sleep (3), REM sleep	Activity monitor ↔ in sleep at night when performing physical work in high (33–35 °C) during day PSG ↓ in light sleep, TST, SOL, and WASO compared to control (*p* < 0.01) ↔ deep sleep with restricted groups compared to control	Sleep restriction alone is more adverse than sleeping in heat.
Dennison et al., 2012 [46]	1-day simulated fire ground test (SFGT): In non-fatigued state or fatigued state after exercises session	Acute fatigue	Blood lactate Heart rate Rating of perceived exertion Total SFGT time	Blood lactate ↔ between condition (*p* > 0.771) Heart rate ↔ between groups (*p* > 0.457) RPE ↓ non-fatigued 8.2 vs. ↑ exercise 9.5 (15% difference) SFGT time ↓ time non-fatigued group 365.0 ± 56.4 s ↑ time exercise group 399.9 ± 70.6 s (*p* < 0.002, effect size 0.546)	Long-term benefits of exercise may outweigh the negatives and those that possess higher fitness levels tend to perform more efficiently.
Ferguson et al., 2016 [49]	3-day 12-h shift simulation with normal sleep with or without hot room or sleep restricted with or – hot room with physical tasks: Weighted tire drag, raking debris, walking with weighted hose while avoiding obstacles, holding a weighted hose rake in static position, and a 25 m fire hose rolling up to operational standard	Acute fatigue	Go/No-Go, Stroop Colour Word Test, and the Occupational Safety Performance Assessment Test Psychomotor Vigilance Task (PVT) Samn–Perelli Fatigue Scale (SPFS) Visual analogue scale (VAS) alertness pre-performance of task motivation	PVT ↓ scores compared to baseline Sleep restricted/cool (β = −0.43, *p* < 0.001) Sleep restricted/hot Sleep restricted/cool (β = −0.63, *p* < 0.001) SPFS ↑ variance in both baseline vs. test (r^2^ 0.60) and change to test with the recovery period (r^2^ 0.70). SPFS compared to VAS (*p* < 0.001, β = 0.90) VAS changes baseline to test and changes test to recovery Alertness (r^2^ 0.53)/r^2^ 0.52) Per-performance (r^2^ 0.49)/(r^2^ 0.49) Motivation (r^2^ 0.42)/(r^2^ 0.44) Other cognitive measures ↔ with other cognitive measures	The PVT was most sensitive objective measure with the SPFS being stronger than self-rated measures of fatigue.
Games et al., 2020 [51]	1-event of the Functional Agility Short-Term Fatigue Protocol	Acute fatigue	Static and dynamic balance: double leg velocity sway single leg sway anterior Y-balance test	Post activity ↑ double-legged displacement (mean difference = 1.3 + 2.8 cm^2^ 95% CI = 0.4, 2.2 cmi d = 0.46: *p* = 0.007) ↑ single-legged sway (mean difference = 2.3 + 4.5 cm^2^ 95% CI = 0.8. 3.8 cm^2^; d = 0.51; *p* = 0.004) ↑ average displacement velocity post activity during double-legged (mean difference = 0.18 + 0.21 cm/s 95% CI = 0.1, 0.3 cm/s; d = 0.85: *p* < 0.001) ↓ anterior reach (mean difference = −1.5 + 2.9 cm; 95% CI = −2.5, −0.6 cm; d= 0.5; *p* = 0.003) in Y-balance	Short bouts of physical exertion negatively affected balance.
Gerstner et al., 2022 [52]	3–24-h shift cycles	Acute fatigue	Actigraphy Reactive isometric force (milliseconds): 50 ms 100 ms 150 ms 200 ms Psychomotor Vigilance Task (PVT)	Actigraphy ↔ sleep patterns from on days compared to off days Reactive isometric force ↓ absolute force at 50 ms Pre: 37.67 ± 42.35; post: 27.90 ± 25.24, mean change −10.28 (CI −19.57, −0.99) (*p* < 0.05) ↔ in absolute reactive force in 100, 150, or 200 ms PVT ↔ PVT Pre: 277.74 ± 52.60; post: 278.40 ± 45.23, mean change 0.65, (CI –18.59, 19.90)	Rapid early force production in 50 ms was decreased on the day off following the common 3–24-h on-off shift cycle.
Giuliani et al., 2020 [54]	Shift cycle: 3-days on/4 days off	Acute fatigue	Body mass index (BMI) Isometric knee extension Occupational Fatigue Exhaustion Recovery Scale (OFERS)	OFERS ↑ age was related to ↑ acute fatigue and chronic fatigue (r=0.545 to 0.551, p=0.001) with ↓ inter-shift recovery (r = −0.448, *p* < 0.01) Knee extension ↔ maximal knee extension strength or BMI with fatigue	Increasing age was associated with poorer recovery between shifts and with increased acute and chronic fatigue.
Kaikkonen et al., 2017 [59]	2–24-h, 6-h ambulance, and 6-h fire and rescue shifts	Acute fatigue	Energy expenditure Heart rate Mean and peak Heart rate variability Oxygen uptake (VO_2_ max) Mean and peak Stress and recovery index	Energy expenditure ↑ 24-h mean calorie expenditure was 2677 ± 658 kcal vs. to 6-h rescue (823 ± 367) or 6-h ambulance (723 ± 232) (*p* < 0.05) Mean and Peak HR ↑ 6-h rescue mean HR (78 ± 12) vs. 24-h (73 ± 7) and 6-h ambulance (71 ± 9) (*p* < 0.001) ↑ 24-h peak HR (156 ± 16) vs. 6-h rescue (136 ± 25) and 6-h ambulance (120 ± 14) (*p* < 0.001) HRV ↓ RMSSD in 6-h rescue (38 ± 16) vs. 24-h (42 ± 14) and 6-h ambulance (45 ± 21) (*p* < 0.01) VO_2_ max (peak; mean) ↑ 24-h (10.6 ± 2.3; 72 ± 11) vs. to 6-h rescue (12 ± 5; 55 ± 19) and 6-h ambulance (9 ± 3; 41 ± 12) (*p* < 0.001) Stress and recovery ↑ stress in 6-h rescue (118 ± 40) vs. 24-h (108 ± 33) or 6-h ambulance (105 ± 36) (*p* < 0.01) ↓ recovery in 6-h rescue (12 ± 14) vs. 24-h (27 ± 11) or 6-h ambulance (28 ± 25) (*p* < 0.01)	High physiological and psychological stress loads were seen in 24-h shifts compared to shorter shifts.
Khan et al., 2021 [61]	2-day shift, 2-night shift, 4-days off: Times measured were pre-shift, standard day shift, nightshift, day off one and two	Acute fatigue	Actiwatch-2: total sleep time (TST), wake after sleep onset (WASO), time in bed (TIB), number of awakenings (NOA), sleep efficiency (SE), sleep latency (SL) Galvanic skin response Karolinska Sleepiness Scale (KSS) Pittsburgh Sleep Diary (Not reported statistically) Positive and Negative Affect Score (PANAS) Samn–Perelli Fatigue Scale (SPFS)	Galvanic response ↔ among the five time points (*p* > 0.05) KSS Stress significantly differed among the rotating shift Before-work level during work days or morning level during non-work days [F(2.78, 22.21) = 8.21, *p* < 0.05; η^2^ = 0.45] During-work levels on work days or afternoon levels during non-work days [F(2.92, 23.35) = 8.43, *p* < 0.05; η^2^ = 0.44] After-work levels during work days or evening levels during non-work days [F(2.19, 17.54) = 16.85, *p* < 0.001; η^2^ = 0.63] NOA NOA during sleep significantly differed within the five time points [F(2.52, 20.14) = 4.736, *p* < 0.05 η^2^ = 0.278] PANAS ↔ among the five time points SE and SL ↔ differences SPFS ↑ fatigue scores during work-on-work days or afternoon levels during non-work days (F(3.10, 24.78) = 8.50, *p* < 0.001; η^2^ = 0.38) ↑ fatigue scores after-work during work days or evenings during non-work days (F(3.18, 25.450) = 20.450, *p* < 0.001; η^2^ = 0.66) TIB TIB differed significantly among the five time points in the shift cycle [F(2.00, 16.01) = 10.18, *p* < 0.05; η^2^ = 0.50] TST TST was significantly different among the five time points in the shift cycle [F(2.06, 22.29) = 12.37, *p* < 0.001; η^2^ = 0.51] WASO WASO differed significantly among the five time points in the rotating shift schedule [F(2.732, 21.85) = 3.93, *p* < 0.05; η^2^ = 0.23]	Levels of fatigue, sleepiness, and stress were all related to the sleep restriction that came with night duty.
Kujawski et al., 2018 [62]	2-day sleep deprivation in laboratory	Acute fatigue	Sprawności Operacyjnej Test: choice reaction delayed matching simple reaction visual attention test	Choice reaction ↑ reaction time (F[5, 270] = 3.63, *p* = 0.003, ω^2^ = 0.02) and in the number of errors committed (ε = 0.90, F[4.50, 242.94] = 4.07, *p* = 0.002, ω^2^ = 0.03) with time spent awake Delayed matching Significant effect of number of committed errors (F[5, 270] = 2.29, *p* = 0.046, ω^2^ = 1.1); This was not observed in the case of correct responses (*p* > 0.05) Simple reaction ↑ in errors on second attempt (ε = 0.80, F[3.99, 215.57] = 3.61, *p* = 0.007, ω^2^ = 0.03) with the first and third attempt showing no significance Visual attention ↑ reaction time (F[5, 270] = 10.59, p, 0.001, ω^2^ = 0.04) ↓ correct responses (F[5, 270] = 9.87, p,0.001, ω^2^ = 0.04)	After 12 h wake cognitive reaction tests had fewer correct responses and increased errors in simple reaction time and peaked at hour 27.
Marcel-Millet et al., 2020 [65]	3-night, 3 experimental conditions: (1) At home (not on shift) (2) At station (no simulation) (3) At station (with simulation: moving two hoses 100 m; (2) obstacle course of 50 m; (3) climbing a 4-storey tower; (4) carry a 60 kg mannequin up/down one floor; (5) going down the 4-storey tower and returning to the starting point	Acute fatigue	Heart rate Heart rate variability Hexoskin sleep measures: total sleep time sleep onset latency sleep efficiency Spiegel Sleep Quality Perception Questionnaire	HR and HRV ↑ effect on condition for HR, mRR, RMSSD, and SD1 (*p* < 0.001) Total sleep time ↓ total sleep (399.5 ± 58.2) regardless of intervention (281.5 ± 67.5) Spiegel score ↓ sleep quality with and without intervention (21 ± 2.9 vs. 18.3 ± 2.1) out of 30, respectively	Being on-call affected autonomic sleep measures regardless of work simulation.
McGillis et al., 2017 [66]	1–7+ day wildfire deployment types: Base work (BW) Initial attack (IA) Project fire (PF)	Acute fatigue	Actigraphy: Total sleep time (TST), wake after sleep on set (WASO), sleep efficiency (SE) Fatigue questionnaire Psychomotor Vigilance Task (PVT)	Fatigue questionnaire ↑ fatigue levels for IA compared to base (X^2^(2) = 10.054, *p* < 0.006) PVT ↑$\;\mathrm{reaction}\;\mathrm{time}\;\mathrm{for}\;\mathrm{mornings}\;\mathrm{for}\;\mathrm{IA}\;(n=6,\;424.8\;\pm \;51.3\;\mathrm{ms})\;\mathrm{compared}\;\mathrm{to}\;\mathrm{PF}\;(n=66,\;372.4\;\pm \;51.1\;\mathrm{ms})\;(p=0.014)\;(X^{2}(2)=8.097,\;p=0.017)\;\mathrm{with}\;↔$ in base scores (n = 19, 385.7 ± 64.2 ms) ↔ in evening reaction time for all conditions SE (<85%) Base 85.7 ± 8 (50%) Initial attack 75.6 ± 19.2 (60%) Project fire 87.6 ± 7.9 (33%) TST (min) (<7-h sleep) Base 371.6 ± 58.1 (87%) Initial attack 287.2 ± 69.3 (100%) Project fire 373.4 ± 55.1 (81%) WASO (>31 min) Base 58.8 ± 33.9 (75%) Initial attack 92.8 ± 82.8 (86.6%) Project fire 51.4 ± 33.7 (68.8%)	Sleep quality and quantity measures were outside of the recommended thresholds in all deployment types.
Nowak and Łukomska, 2021 [67]	Multiple days live job assessment: 24-h shift firefighters; 12-h shift paramedics; 8-h shift controls (office workers)	Acute fatigue	Colour Trails Test (CTT) Perceived workload: Physical and psychological D2 test Epworth Sleep Scale (ESS) Health Behaviour Inventory (HBI) Karolinska Sleepiness Scale (KSS) Pittsburgh Sleep Quality Index (PSQI)	CTT Workload ↔ in perceived mental workload between groups ↑ perceived physical workload in firefighters and paramedics vs. controls (H(2) = 21.226, *p* < 0.001) ESS ↔ seen in any scores between groups D2 test and CTT ↔ in group differences HBI ↓ health behaviour in paramedics compared to firefighters (*p* < 0.032) KSS ↔ when compared to both cognitive test ↔ in scores for both firefighters and control group ↑ sleepiness in paramedics after night shift (Mdn 6.5) vs. after day shift (Mdn 4; *p* < 0.014) and on day off (Mdn4; *p* < 0.001) PSQI ↓ mean average sleep for paramedics (5.75 h) vs. firefighters (7 h; *p* < 0.0016) and controls (7 h; *p* < 0.001)	Paramedics were most affected by shift work in sleep quality, duration, and decreases in general health scores.
Pau et al., 2014 [72]	1-event of firefighter specific simulated tasks	Acute fatigue	Centre of pressure in postural balance in career vs. retained firefighters	↔ in pre-activity measures in career or retained firefighters Pre- to post-activity ↓ career (medial lateral scores = −0.23 (−1.11, 0.64) ↑ retained FFs (medial lateral scores = −1.59 (−2.35, −0.83)	Retained firefighters have more risk of balance-related injuries than career firefighters.
Smith et al., 2016 [75]	3-day 12-h shift simulation with normal sleep, normal sleep + hot, sleep restricted, or sleep restricted + hot: Weighted tire drag, raking debris, walking with weighted hose while avoiding obstacles, holding a weighted hose rake in static position, and a 25 m fire hose rolling up to operational standard	Acute fatigue	Psychomotor Vigilance Task (PVT) Visual analogue scale (VAS): self-perceived performance	PVT vs. VAS ↑ mean reaction time on PVT ↓ in predicting their own performance with the VAS (r = −0.61, X^2^ (1) = 19.1, *p* < 0.001)	The ability to predict fatigue lessened with each day becoming less reliable.
Vincent et al., 2015 [82]	4-day simulation of six firefighter specific tasks—sleep restricted (SR) vs. control: Charged hose advance, blackout hose work, hose rolling, lateral repositioning, rake, and static hold	Acute fatigue	Core temperature Heart rate Polysomnography (PSG) Rating of perceived exertion (RPE)	Core temperature and heart rate ↔ with sleep restriction PSG ↓ mean sleep duration in SR group (3.6 ± 0.3 h) compared to control group (6.9 ± 0.4 h) (*p* < 0.001) RPE ↔ between SR and control group	Sleep restricted firefighters’ physical performance was largely unaffected by 4-h of sleep.
Vincent et al., 2016 [79]	2–9-day wildfire deployment	Acute fatigue	Actigraphy: total sleep time (TST), sleep efficiency (SE), sleep latency (SL), sleep quality SQ), time woken (TW) Samn–Perelli Fatigue Scale (SPFS) Sleep diary/work diary Sleep location	TST (hours) Non-fire day (7.0 ± 0.9) and fire day (6.1 ± 1.7) (*p* < 0.001) ↔ between for SE, SL, TW, or subjective sleep quality on non-fire and fire days. SPFS ↑ fatigue pre-sleep compared to post sleep on both fire days (1.17 ± 0.17) and non-fire (1.24 ± 0.18) days (*p* < 0.001) Sleep location ↓ total sleep time when sleeping in tent or vehicle compared to motel or home (*p* < 0.01) No statistical significance in sleep diary and work diary had	Sleep location, shift length and shift start times have the potential to be areas to focus on to improve sleep quality and should be identified in future fatigue risk management strategies.
Vincent et al., 2017 [80]	3-day 10-h shift simulation with sleep restriction or hot + sleep restriction: Charged hose advance, blackout hose work, hose rolling, lateral repositioning, rake, and static hose hold.	Acute fatigue	Core temperature Heart rate Polysomnography (PSG) Rating of Perceived Exertion (RPE) Task performance	Heart rate and core temperature ↔ in either outcomes PSG ↔ in either group sleep duration RPE ↑ increase as simulation went on regardless of condition Task performance Sleep restricted/hot group covered less area than sleep restricted by 10–40 m (*p* < 0.001)	Sleep restriction with heat did not differ in physiological responses, motivation or RPE.
Vincent et al., 2018 [81]	3-day 10-h shift simulation with hot normal sleep or hot sleep restriction: Charged hose advance, blackout hose work, hose rolling, lateral repositioning, rake, and static hose hold.	Acute fatigue	Heart rate Polysomnography (PSG) Rating of Perceived Exertion (RPE) Work performance	PSG ↓ HOT + SR group (3.5 ± 0.5 h) compared to HOT (6.7 ± 0.9 h; *p* < 0.001) RPE ↑ Hose rolling (β = 0.87 ± 0.39; *p* = 0.027) and static hold (β = 1.51 ± 0.70; *p* = 0.031) Task performance and heart rate indicate significant inter-individual variability independent of condition	Physical performance was not impacted by sleep restriction.
Courtney et al., 2013 [41]	Survey	Chronic fatigue	Depression Anxiety Stress Scale-21 (DASS21) International Physical Activity Questionnaire-Short Form (IPAQ) Pittsburgh Sleep Quality Index (PSQI) Standard Shiftwork Index- Chronic Fatigue Scale (SSI-CFS)	DASS21 and PSQI ↑ chronic fatigue predicted by sleep quality (β = 0.43, *p* < 0.001) depression (β = 0.25, *p* < 0.03) Remaining variable did not significantly contribute: total METs, β = −0.08, *p* = 0.26, stress, β = 0.0.09, *p* = 0.46, and anxiety β = 0.03, *p* = 0.81	The largest predictor of chronic fatigue was lack of sleep.
Courtney et al., 2020 [42]	Survey	Chronic fatigue	Ecological Momentary Assessment (EMA) Visual Analogue Scale: stress tiredness	EMA-acute stress ↓ acute stress when “off night/day” (β_1_ =16.27) ↑ levels of acute stress when “on night/day” (β_1_ + β_2_ =24.47) ↑ sleep disruptions ↑ stress by 0.65 points on VAS (β_5_, *p* < 0.001) EMA-acute tiredness ↓ acute tiredness ‘‘off night/day’’ (β_0_ =24.68) ↑ acute tiredness when “on night/day” (β_0_ + β_1_ =30.00) ↑ sleep disruptions ↑ tiredness by 1.743 points on VAS (β_4_, *p* < 0.011) ↓ acute tiredness from taking nap by 2.670 points on VAS (β_5_, *p* < 0.027)	Sleep disruptions contributed to increased levels of both stress and tiredness.
Dawson et al., 2015 [44]	Interview	Chronic fatigue	Open-ended questions based on perceptions, attitudes and experience of safety, opinions, and fatigue management systems with the organisation	Qualitative synthesis: Areas identified: physical manifestations, cognitive effects, loss of situational awareness, memory effects, difficulties with decision making, and communication problems	Fatigue was a clear aspect in all volunteer firefighters and led to safety issues.
de Carvalho Dutra, 2017 [45]	Survey	Chronic fatigue	Habitual Physical Activity Questionnaire (AFH) Bipolar Fatigue Evaluation Questionnaire (BFEQ) Pittsburgh Sleep Quality Index Scale of Stress at Work	AFH 75% performed physical activity at least ×1/week BFEQ 70% reported intense fatigue and 30% reported moderate fatigue at end of shift PSQI 60% reported poor sleep with 15% having sleep disturbances Stress 65% reported moderate stress, 15% high stress, and 20% mild stress	Fatigue was reported by all respondents with physical or mental tiredness.
Donnelly et al., 2019 [47]	Survey	Chronic fatigue	Chalder Fatigue Questionnaire Emergency Medical Services Safety Inventory (EMS-SI) injury, adverse events, safety-compromising behaviours	Adverse events ↑ when fatigued (β 0.41, OR 1.50, 95% CI 1.06–2.11) ↑ with shift length 12+ hours (β 1.13, OR 3.01, 95% CI 1.31–7.26) ↑ with age (β −0.04, OR 0.958, 95% CI 0.94–0.97) Injury ↑ when fatigued (β 0.76, OR 2.13, 95% CI 1.54–2.96) ↑ with rotating shift (β 0.35, OR 1.42, 95% CI 0.96–2.08) Safety-compromising behaviours ↑ when fatigued (β 1.19, OR 3.27, 95% CI 1.27–8.47) ↑ with 40+ hours worked (β 1.78, OR 5.90, 95% CI 2.54–13.74)	Those working greater than 40 h a week display were increased fatigue and had a decrease in safety outcomes.
Donnelly et al., 2020 [48]	Survey	Chronic fatigue	Chalder Fatigue Questionnaire (CFQ) Emergency Medical Services Chronic Stress Questionnaire Operational stress, organisational stress, critical incident stress Emergency Medical Services Safety Inventory (EMS-SI): injury, adverse events, safety-compromising behaviours PTSD Checklist- Military	Injuries ↑ weak to moderate (r = 0.26–0.40) positive correlation to organisational stress, operational stress, critical incident stress, PTSD, and fatigue Safety-comprising behaviours ↑ weak to moderate (r = 0.30–0.39) positive correlation to organisational stress, operational stress, critical incident stress, PTSD, and fatigue	Fatigue was significantly related to all stress factors and associated with safety compromising behaviours, and injuries/exposures.
Fullagar et al., 2021 [50]	Survey	Chronic fatigue	Visual analogue scale of mental and physical fatigue:	VAS ↑ fatigue during average task, mental fatigue 4.2 ± 2.4, physical fatigue 4.9 ± 2.4 ↑ fatigue during demanding task, mental fatigue 6.7 ± 2.2, physical fatigue and 7.6 ± 1.8 (Scores were not compared statistically)	The most mentally demanding tasks reported were rescue, structural firefighting, and bushfire-fighting.
Ghasemi et al., 2021 [53]	Survey	Chronic fatigue	Multidimension Fatigue Inventory (MFI) Perceived safety climate questions Safety behaviour items from the NFPA 1500	Fatigue ↓ safety behaviour (r = −0.32, *p* < 0.01) ↓ safety climate (r = −0.39, *p* < 0.01)	Fatigue negatively affects safety behaviour.
Huang et al., 2022 [55]	Survey	Chronic fatigue	Pittsburgh Sleep Quality Index (PSQI)	PSQI ↑ scores among 31–45 yr compared to <30 yr (β = −1.13, *p* < 0.05) and >45 yr (β = −0.92, *p* > 0.05) ↑ PSQI with 48 on/24 off compared to 24 on/24 off (OR 0.60 (95% CI 0.43, 0.84) (*p* < 0.01)	Those with working longer shift cycles had worse sleep quality.
Jeklin et al., 2020 [56]	17-day fire line deployment (14-day work with 3-day rest)	Chronic fatigue	Actigraphy: total sleep time (TST), wake after sleep on set (WASO), sleep latency (SL), sleep efficiency (SE) Psychomotor Vigilance Task (PVT) Sleep diary (Used to enhance actigraphy) Visual analogue scale (VAS 0–10 cm): fatigue alertness sleepiness	PVT ↑ reaction time Day 13 (267.1 ± 32 msec) vs. day 5 (253.4 ± 29.7 msec) (*p* = 0.025) TST, WASO, SE, and SL ↔ between fire and non-fire days for VAS of fatigue ↑ fatigue day 13 (6.0 ± 1.9 cm) vs. day 3 (4.2 ± 2.2 cm) (*p* = 0.033) ↑ fatigue day 16 (M = 6.3 ± 2.2 cm) vs. day 5 (4.1 ± 2.2 cm), (*p* = 0.025) VAS of alertness ↓ alertness day 3 (6.5 ± 1.2 cm) vs. today 13 (4.7 ± 1.8 cm) (*p* = 0.003) VAS of sleepiness ↑ sleepiness day 16 (6.5 ± 2.3 cm) vs. day 1 (4.3 ± 2.2 cm) (*p* = 0.038)	As deployment length increased so did objective and subjective fatigue measures.
Jeklin, Davies, et al., 2021 [83]	17-day wildfire deployment (14-day work with 3-day rest)	Chronic fatigue	Circadian Alertness Simulator	The range of scores for the circadian alertness simulator was from 21.6–56.3 (29.4 ± 6.2) with no firefighters having risk ↑ scores (>60)	All reported some levels of fatigue but none were high risk of accidents.
Jeklin, Perrotta, et al., 2021 [57]	14-day wildfire deployment	Chronic fatigue	Actigraphy: total sleep time (TST) Heart rate variability (HRV) Reaction time: simple, choice, and discrimination reaction time Visual Analogue Scale (VAS 0–10 cm): fatigue, alertness, and sleepiness	HRV vs. ↑ sleepiness Ln rMSSD (r = −0.60, *p* = 0.000) ↑ fatigue Ln rMSSD (r = −0.55, *p* = 0.000) ↓ total sleep time Ln rMSSD (r = 0.28, *p* = 0.009) Total sleep time (min) compared to controls ↓ Day 1 (377.7 ± 32.6) (−13.5%), day 3 (378.9 ± 30.5) (−13.2%), and day 12 (356.1 ± 53.7) (−18.4%) (*p* < 0.003) VAS of fatigue ↑ Day 1 (3.1 ± 2.1) vs. day 13 (6.2 ± 1.9) (94.6%) (*p* < 0.004) VAS of sleepiness ↑ Day 1 (2.8 ± 2.5) vs. day 11 (5.7 ± 2.2) (105.2%), day 13 (6.1 ± 1.9) (119.9%), and day 14 (6.3 ± 1.8) (124.7%) (*p* < 0.004) VAS of alertness ↓ Day 1 (7.2 ± 1.5) vs. day 11 (4.4 ± 1.8) (−39.3%), and day 13 (3.8 ± 1.8) (−47.4%) (*p* < 0.004) Reaction compared to HRV No significant differences	HRV was significantly associated to increased age, subjective ratings of fatigue, and alertness as deployment time increased.
Khan et al., 2020 [60]	Survey	Chronic fatigue	Beck Depression Inventory-Short Form Berlin Questionnaire for OSA Bruxism Assessment Questionnaire Epworth Sleepiness Scale Fatigue Severity Scale General Health Questionnaire (SF-36) Insomnia Severity Index Perceived Stress Scale Pittsburgh Sleep Quality Index Pittsburgh Sleep Quality Index-Addendum for PTSD Shift-work Disorder Screening Questionnaire State-Trait Anxiety Inventory-Short Form Ullanlinna Narcolepsy Scale	Depression and Anxiety ↑ strong negative (r = −0.70 to −1) correlation to mental health ↑ moderate negative (r = −0.50 to −0.70) correlation to role emotional, social functioning, vitality, and general health ↑ moderate positive (r = 0.50–0.70) correlation to insomnia, PTSD, and sleep quality	Paramedics have a high prevalence of sleep quality, insomnia, and mental health issues.
Lin et al., 2020 [64]	Survey	Chronic fatigue	Emergency Medical Services Safety Inventory (EMS-SI) Epworth Sleepiness Scale (ESS) Workload questions	ESS vs. EMS-SI ↑ mild sleepiness (ESS score 8~11, 36.9%)=↑ in injury score ×0.173 (*p* < 0.05) ↑ excessive sleepiness (ESS score ≧12, 39.2%)=↑ injury score ×0.193 (*p* < 0.05) Workload and injury No significant relationship found	Sleepiness is a key risk factor in EMS for safety and injury issues.
Paterson et al., 2014 [69]	Survey (open answer)	Chronic fatigue	“What do you believe your fatigue is a result of?”	Qualitative synthesis: Six themes were identified: Working time, sleep, workload, health and wellbeing, work–life balance, and environment	Major contributors to fatigue were reported as nightshift, inadequate rest/breaks, insufficient sleep, sleep difficulties, and high/excessive workload.
Paterson et al., 2016 [68]	Interview	Chronic fatigue	Factors increasing health and safety risk	Qualitative synthesis: Factors related to health and safety risk were fatigue and sleep and caused by sleep disruption, expectation of an alarm, fatigue and driving after waking were identified as risk associated to sleep and fatigue	Fatigue is a significant issue for firefighters with retained firefighters reporting higher levels of fatigue.
Patterson et al., 2016 [70]	Survey	Chronic fatigue	Chalder Fatigue Questionnaire Epworth Sleepiness Scale Occupation Fatigue Exhaustion Recovery Scale (OFERS) Pittsburgh Sleep Quality Index Sleep Fatigue and Alertness Behaviour	All measures ↑ fatigue, sleepiness, and concentration in a 24-h shift vs. 8-h shift (*p* < 0.05)	Changing from 24-h shift to 8-h shift substantially improved overall fatigue levels.
Patterson et al., 2012 [71]	Survey	Chronic fatigue	Chalder Fatigue Questionnaire (CFQ) EMS Safety Inventory (EMS-SI) Pittsburgh Sleep Quality Index (PSQI)	CFS ↑ fatigued while at work (n = 281, 55.0%; 95% CI 50.7, 59.3%) EMS-SI ↑ injury ×2.9 rates in fatigued vs. non-fatigued (OR = 2.9, 95% CI 1.8, 4.6) ↑ error/adverse events ×2.3 in fatigued vs. non-fatigued (OR = 2.3, 95% CI 1.5, 3.3) ↑ compromised safety ×4.9 in fatigued vs. non-fatigued (OR = 4.9, 95% CI 2.4, 9.8) PSQI ↓ sleep quality (n = 304, 59.5%; 95% CI 55.2–63.8%)	Fatigue and poor sleep can increase injury and decrease safety outcomes in provider and patient.
Pyper and Paterson, 2016 [73]	Survey	Chronic fatigue	Chalder Fatigue Questionnaire (CFQ) Depression Anxiety Stress Scale (DASS-21) Impact Event Scale	Descriptive analytics was used ↑ levels of fatigue and emotional trauma in rural and regional paramedics	Ambulance personnel have increased experiences of stress, fatigue, and emotional trauma.
Rodríguez-Marroyo et al., 2012 [74]	4 consecutive wildfire seasons (Average 15 fire/subject)	Chronic fatigue	Core body temperature Cumulative Heat Strain Index (CHSI) Exercise workload (TRIMP) Heart rate Physiological Strain Index (PSI)	Core body temperature and heart rate ↔ in scores throughout seasons TRIMP ↑ score with wildfire duration (*p* < 0.05) TRIMP vs. CHSI (r = 0.88, *p* < 0.001) CHSI ↑ cardiovascular and thermal stress as duration increased (*p* < 0.05) PSI ↔ scores were similar during all wildfires	Heart rate and core temperature were not reflective of thermal or cardiovascular strain during wildfire deployment.
Sofianopoulos et al., 2011 [76]	Survey	Chronic fatigue	Beck depression inventory Berlin Questionnaire Epworth Sleepiness Scale (Scores were not compared statistically) Pittsburgh Sleep Quality Index	PSQI vs. fatigue (r = −0.459, n= 59, *p* = 0.000) No correlation was found for fatigue and the other variables	Paramedics report poor sleep quality, fatigue, and performing at suboptimal levels.
Toyokuni et al., 2022 [77]	Survey	Chronic fatigue	“During the past month, how has your fatigue level been after working?” 5-point Likert scale “During emergency rescues in the past month, have you experienced near-miss incidents?”	↑ high or very high levels of fatigue was associated with ↑ near-misses (OR 3.19, 95% (CI): 1.68–6.05)	Fatigue combined with an unhealthy lifestyle was associated to greater near-miss incidents.
Vincent et al., 2016 [78]	4-weeks of planned burns	Chronic fatigue	Actigraphy: time in bed (TIB), total sleep time (TST), sleep efficiency (SE), sleep latency (SL) Samn–Perelli Fatigue Scale (SPFS) Sleep diaries Sleep location	Sleep quality ↔ in total sleep time, time in bed, sleep efficiency, sleep latency or subjective sleep duration, times woken, sleep quality between non-burn and burn days SPFS ↑ fatigue pre-sleep vs. post-sleep on non-burn (0.80 ± 0.19) and burn (0.9 ± 0.18) days (*p* < 0.001) ↑ fatigue on burn days were higher vs. non-burn days in pre-sleep (0.3 ± 0.18; *p* = 0.001) and post-sleep (0.2 ± 0.19; *p* = 0.004) Sleep location ↔ in sleep location between non-burn and burn days	Sleep quality and quantity are not affected unless shifts are >12 h.
Jeong et al., 2019 [58]	Shift cycle vs. day only + 1 rest day: 3- day 6-day 9-day 21-day	Combined	Actigraphy: total sleep time (TST), time in bed (TIB), sleep latency (SL), sleep efficiency (SE), wake after sleep onset (WASO)	Actigraphy ↑ sleep latency (10.8 ± 3.8, 12.6 ± 6.9) and wake after sleep onset (53.5± 24.8, 78.3± 40.6) in day only vs. shift work, respectively (*p* < 0.05) ↓ total sleep time in shift work (266.9 ± 84.8) compared to day only (347.7 ± 87.6) (*p* < 0.05) ↑ sleep efficiency in 6-day shift compared to other shifts (*p* < 0.05) ↓ sleep efficiency in 21-day shift compared to other shifts (*p* < 0.05)	Sleep quality on night shift and on rest day were lower than controls.
Kwak et al., 2020 [63]	Shift cycle: 3-day 6-day 9-day 21-day	Combined	Central Nervous System Vital Signs (CNSVS) Insomnia Severity Index (ISI) The Patient Health Questionnaire-9 (PHQ-9)	CNSVS day vs. night work ↓ composite memory (90.6 ± 19.1 vs. 84.7 ± 19.7) (*p* < 0.001) ↓ verbal memory (87.7 ± 20.0 vs. 81.3 ± 21.9) (*p* < 0.001), ↓ visual memory 97.1 ± 16.3 vs. 94.0 ± 16.6 (*p* < 0.001), ↓ psychomotor speed (112.4 ± 15.4 vs. 110.1 ± 15.2) (*p* < 0.001) ↓ motor speed (111.0 ± 15.1 vs. 108.7 ± 14.1) (*p* < 0.001) ↓ complex attention (97.8 ± 18.2 vs. 93.3 ± 32.4) (*p* < 0.007) CNSVS vs. those with mild insomnia after night work ↓ composite memory (90.6 ± 17.2 vs. 85.4 ± 19.4) (*p* = 0.002) ↓ verbal memory (87.4 ± 19.1 vs. 80.4 ± 20.3) (*p* < 0.001) ↓ complex attention (100.1 ± 15.0 vs. 92.2 ± 43.3) (*p* < 0.027) ↓ psychomotor speed (114.1 ± 14.5 vs. 111.5 ± 14.5) (*p* < 0.008) ↓ motor speed (112.2 ± 14.8 vs. 108.3 ± 13.5) (*p* = 0.001) CNSVS and those with insomnia after day vs. night work ↓ composite memory (92.8 ± 21.6 vs. 81.5 ± 21.9) (*p* = 0.012) ↓ verbal memory (89.7 ± 20.1 vs. 77.2 ± 24.6) (*p* = 0.001) ↓ motor speed (111.4 ± 12.1 vs. 104.8 ± 13.6) (*p* = 0.007) PHQ-9 and those with depression after day vs. night work ↓ verbal memory 90.1 ± 20.9 vs. 82.7 ± 21.3 (*p* < 0.001) ↓ psychomotor speed 112.9 ± 13.5 vs. 105.4 ± 15.9 (*p* < 0.001) ↓ motor speed 113.2 ± 12.7 vs. 105.8 ± 14.3 (*p* < 0.001)	Regardless of shift cycle neurocognitive function showed significant decrease after night shift.

↑ = increase; ↓ = decrease; ↔ = no change; *p* = *p*-values; RR = risk ratios; CI = confidence intervals; SMD = standardised mean differences; r and r^2^ = correlation coefficients; β = beta values; ω^2^ = omega squared; F = F-values; η^2^ = eta squared values; lnRMSSD log-transformed root mean square of successive R-R intervals; X^2^(2) = chi-squared distributions, OR = odds ratios; BFEQ = Bipolar Fatigue Evaluation Questionnaire; PSI = physiological strain; CHSI = cumulative heat strain; TRIMP = exercise workload; SPFS = Samn–Perelli fatigue scale; PVT = psychomotor vigilance task; VAS = visual analogue scale; HRV = heart rate variability; TST = total sleep time, WASO = wake after sleep onset; SL = sleep latency, SE = sleep efficiency; NOA = number of awakening; PSG = polysomnography; CNSVS = central nervous system vital signs; ISI = insomnia severity index; PHQ-9 = patient health questionnaire; OFERS = occupational fatigue exhaustion recovery scale; BMI = body mass index; KSS = Karolinska sleepiness scale; PANAS = positive and negative affect scale; ESS = Epworth sleepiness scale; CFS = Chalder fatigue scale; PSQI = Pittsburgh sleep quality index; DASS-21 = depression anxiety stress scale-21; IPAQ= international physical activity questionnaire-short form; EMS-SI; emergency medical service safety inventory; HBI = health behaviour inventory; CTT = colour trails test; SFGT = simulated fire ground test; MFI = multidimension fatigue inventory; EMA = ecological momentary assessment.

## Data Availability

Data is contained within the article. Preregistration was filed in OSF: osf.io/26f3s.

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
