# Peer review of "Occupation-Induced Fatigue and Impacts on Emergency First Responders: A Systematic Review"

_ijerph, 2023, doi:10.3390/ijerph20227055_

Round 1
Reviewer 1 Report
Comments and Suggestions for Authors
Dear athours
There are a few comments:
1. The Joanna Briggs Institute (JBI) at the top of Table 3 is to be presented thoroughly.
2. Two boxes in Figure 1 on the right side have unclear writings that need to be corrected.
3. I think the text of the manuscript needs minor writing editing.
Sincerely
Comments on the Quality of English LanguageDear authors
There are a few comments:
1. The Joanna Briggs Institute (JBI) at the top of Table 3 is to be presented thoroughly.
2. Two boxes in Figure 1 on the right side have unclear writings that need to be corrected.
3. I think the text of the manuscript needs minor writing editing.
Sincerely
Author Response
Reviewer 1
The Joanna Briggs Institute (JBI) at the top of Table 3 is to be presented thoroughly.
- Thank you for your comment. The Joanna Briggs Institute has been spelled out in its entirety.
Two boxes in Figure 1 on the right side have unclear writings that need to be corrected.
- Thank you for your comment. The boxes have been correct. This appears to be a conversion error.
I think the text of the manuscript needs minor writing editing.
- Thank you for you feedback. We have adjusted the manuscript.
Reviewer 2 Report
Comments and Suggestions for Authors
General Comments:
The manuscript provides a comprehensive review of relevant literature regarding the impact of fatigue on first responders' cognitive and physical outcomes. The manuscript is well written, timely, and will have a high impact in the field.
Specific Comments
Pg. 2: The following statements should be revised as the economic cost of firefighter injuries due to fatigue is likely unknown. I believe the following statement estimates the cost of all musculoskeletal injuries, that may not be exclusive of fatigue-induced mechanisms. "Unfortunately, injuries due to occupational fatigue are problematic worldwide. In United States firefighters, the economic cost ranges from $1.6 -$5.9 billion annually [22].
Pg 8: Ensure all text is readable in the Figure 1. Some is cut-off. Spell out PRISMA.
Table 2: Ensure "Firefighter" is plural for some references, as appropriate. -ie, Fullagar et al., Marcel-Millet et al., Smith et al., etc.
Table 3: Describe all questions. Spell out abbreviations in table / figure titles.
Table 4: Provide a more descriptive title. Recommend describing / categorizing each study based on assessment of acute vs. chronic fatigue model...consider placing chronic fatigue studies first, then acute fatigue studies.
Pg 31: Revise Dennison et al. reference to "group" as "condition". It was the same group in different conditions. Also, consider including new references evaluating impact of on-duty high intensity resistance training on subsequent occupational performance a much larger effect than circuit training on multiple performance parameters (air consumption, work efficiency, and work rate)
Pg 33: Some references are not in alphabetical order within a given "environment" category...if that was the intention. Is "Methodology" a more appropriate term to categorize the studies in (vs. Fatiguing environment)?
Pg. 35: Omit first name of author (Sarah) Sofianopoulos et al.
Consider adding updated literature to Discussion section, as appropriate.
Pg 42: To clarify, task completion time increased in the trained group post-exercise. However, the mean completion time post-exercise was still faster than 70% of the untrained, non-fatigued firefighters. -Highlighting the importance of training status (ie, performing regular exercise on- and off- duty).
Pg. 42: Clarify the following statement. 75% achieve recommendation or perform 1 exercise bout per week?
Conversely, 75% of military firefighters reportedly engage in at least one type of moderate to vigorous exercise of at least 150 minutes per week, with 35% engaging in two physical exercises per week [42]
Pg. 46: Are there occupationally relevant measures of cognitive function to report?
Pg. 46: PSQI is commonly used to assess sleep among structural firefighters. However, it may have limitations in determine sleep outcomes of on- vs. off-duty based on the fact that it assesses sleep over the last 30 days. This is important to acknowledge regarding its appropriateness for EFRs.
Pg. 51: 4.7: Consider existing literature on upper body fatigue
Author Response
Reviewer 2
The manuscript provides a comprehensive review of relevant literature regarding the impact of fatigue on first responders' cognitive and physical outcomes. The manuscript is well written, timely, and will have a high impact in the field.
- Thank you very much for your feedback and for reviewing the paper in depth. We are hoping to provide value and insight into the EFR population to aid in injury reduction and improvement in job performance.
Specific Comments
Pg. 2: The following statements should be revised as the economic cost of firefighter injuries due to fatigue is likely unknown. I believe the following statement estimates the cost of all musculoskeletal injuries, that may not be exclusive of fatigue-induced mechanisms. "Unfortunately, injuries due to occupational fatigue are problematic worldwide. In United States firefighters, the economic cost ranges from $1.6 -$5.9 billion annually [22].
- Thank you for your comment. You are correct and that section has been adjusted to only discuss injuries.
Pg 8: Ensure all text is readable in the Figure 1. Some is cut-off. Spell out PRISMA.
- Thank you for your comment. The table has been corrected.
Table 2: Ensure "Firefighter" is plural for some references, as appropriate. -ie, Fullagar et al., Marcel-Millet et al., Smith et al., etc.
- Thank you for your comment. We have corrected the grammar for those studies.
Table 3: Describe all questions. Spell out abbreviations in table / figure titles.
- Thank you for your comment. Spelling out each question would increase the word count exponentially. Alternatively, we have added a statement at the bottom of each table with a link to the corresponding JBI checklist in full.
Table 4: Provide a more descriptive title. Recommend describing / categorizing each study based on assessment of acute vs. chronic fatigue model...consider placing chronic fatigue studies first, then acute fatigue studies.
- Thank you for your comment. We have left the title the same as we feel that it reflects the intent of the paper. However, we have added a column and stated whether each study was acute fatigue, chronic fatigue, or combined and then reordered the studies based in the aforementioned order as this is a natural time-logical progression.
Pg 31: Revise Dennison et al. reference to "group" as "condition". It was the same group in different conditions. Also, consider including new references evaluating impact of on-duty high intensity resistance training on subsequent occupational performance a much larger effect than circuit training on multiple performance parameters (air consumption, work efficiency, and work rate)
- Thank you for your comment and noticing the mistake. We have corrected the term. That reference looks great; however, it was published after the SR search was run. May be good to add in the discussion and to reference going forward.
Pg 33: Some references are not in alphabetical order within a given "environment" category...if that was the intention. Is "Methodology" a more appropriate term to categorize the studies in (vs. Fatiguing environment)?
- Thank you for your comment. After reorganizing the studies based on whether they were acute fatigue, chronic fatigue or combined we also reordered everything else in Table 4 alphabetically. Also, we agree with you that ‘environment’ was the wrong term but ‘methodology’ did not fit either. Instead, we changed it to ‘fatiguing variable’.
Pg. 35: Omit first name of author (Sarah) Sofianopoulos et al.
- Thank you for your comment. We have corrected the mistake.
Consider adding updated literature to Discussion section, as appropriate.
- Thank you for your feedback. We have looked into our discussion and believe there are many current studies added. If there are any studies in particular you are thinking should be updated, please let us know so we can identify and adjust as needed.
Pg 42: To clarify, task completion time increased in the trained group post-exercise. However, the mean completion time post-exercise was still faster than 70% of the untrained, non-fatigued firefighters. -Highlighting the importance of training status (ie, performing regular exercise on- and off- duty).
- Thank you for your comment. That is a great point. We have adjusted that section to highlight that fact and added in another part in the discussion about on-duty exercise to combat physical fatigue.
Pg. 42: Clarify the following statement. 75% achieve recommendation or perform 1 exercise bout per week?
Conversely, 75% of military firefighters reportedly engage in at least one type of moderate to vigorous exercise of at least 150 minutes per week, with 35% engaging in two physical exercises per week [42]
- Thank you for your comment. We have clarified the sentence to state that 75% meet the recommended weekly amount.
Pg. 46: Are there occupationally relevant measures of cognitive function to report?
- Thank you for your comment. There are no occupational measures of cognitive function that exist to our knowledge. The issue is that most cognitive function tests are reaction time and cannot assess situational awareness. The development of one is greatly needed.
Pg. 46: PSQI is commonly used to assess sleep among structural firefighters. However, it may have limitations in determine sleep outcomes of on- vs. off-duty based on the fact that it assesses sleep over the last 30 days. This is important to acknowledge regarding its appropriateness for EFRs.
- Thank you for your feedback. That is a very valid point for those reading to be aware of. We have added a sentence explaining that the PSQI assesses sleep quality over the last 30 days.
Pg. 51: 4.7: Consider existing literature on upper body fatigue
Thank you for your comment. This paper is a part of PhD research and going forward we hope to assess upper body fatigue and lower body fatigue to compare the two and to see how they coexist.
Reviewer 3 Report
Comments and Suggestions for Authors
Presented article with Title Occupation-Induced Fatigue and Impacts on Emergency First Responders: A Systematic Review” is writing on 59 pages with 1 figure, 4 tables and 151 references. The paper interesting, and I think deserves pubblication. I have several comments that need to be resolved.
Comments:
· Abstract is too theoretical.
· The introduction is writing clearly however, but too theoretical. As focused on a large area.
· In article lacks conclusions, what is outputs in this systematic review.
· Check correct formatting of table 4.
· Conclusions/summary, please describe your future directions.
All the specific comments can be followed in reviewed copy of the manuscript.
I recomend this paper publish in journal after minor revision.
Author Response
Reviewer 3
Presented article with Title Occupation-Induced Fatigue and Impacts on Emergency First Responders: A Systematic Review” is writing on 59 pages with 1 figure, 4 tables and 151 references. The paper interesting, and I think deserves publication. I have several comments that need to be resolved.
Comments:
Abstract is too theoretical.
- Thank you for your comment. We have rearranged the abstract to be more of an objective representation of the article.
The introduction is writing clearly however, but too theoretical. As focused on a large area.
- Thank you for your comment. We added more information into the introduction to have more objective numbers about the injury rates from around the world to make it less theoretical.
In article lacks conclusions, what is outputs in this systematic review.
- Thank you for you comment. The conclusion now includes more of a summary with highlights of the main findings along with the most commonly used outcome measures for a practical understanding for the reader.
Check correct formatting of table 4
- Thank you for your comment. The table has been corrected.
Conclusions/summary, please describe your future directions.
- Thank you for your comment. The PRISMA guidelines section 23d state “make explicit recommendations for future research”. Due to the massive future research section, we felt that it warranted a section of its own.
All the specific comments can be followed in reviewed copy of the manuscript.
I recommend this paper publish in journal after minor revision.
- Thank you very much for your time reviewing the paper and suggestions.